



# Two decades observing smoke above clouds in the south-eastern Atlantic Ocean: Deep Blue algorithm updates and validation with ORACLES field campaign data

Andrew M. Sayer[1,2], N. Christina Hsu[2], Jaehwa Lee[2,3], Woogyung V. Kim[2,3], Sharon Burton[4], Marta A. Fenn[4,5], Richard A. Ferrare[4], Meloë Kacenelenbogen[6,7], Samuel LeBlanc[6,7], Kristina Pistone[6,7], Jens Redemann[8], Michal Segal-Rozenhaimer[6,7], Yohei Shinozuka[6,9], and Si-Chee Tsay[2]

[1]GESTAR, Universities Space Research Association, Columbia, MD, USA
[2]NASA Goddard Space Flight Center, Greenbelt, MD, USA
[3]University of Maryland, College Park, MD, USA
[4]NASA Langley Research Center, Hampton, VA, USA
[5]Science Systems and Applications, Inc, Hampton, VA, USA
[6]Bay Area Environmental Research Institute, Moffett Field, CA, USA
[7]NASA Ames Research Center, Moffett Field, CA, USA
[8]University of Oklahoma, Norman, OK, USA
[9]Universities Space Research Association, Mountain View, CA, USA

**Correspondence:** Andrew M. Sayer (andrew.sayer@nasa.gov)

**Abstract.**

This study presents and evaluates an updated algorithm for quantification of absorbing aerosols above clouds (AACs) from passive satellite measurements. The focus is biomass burning in the south-eastern Atlantic Ocean during the 2016 and 2017 ObserRvations of Aerosols above CLouds and their interactionS (ORACLES) field campaign deployments. The algorithm retrieves the above-cloud aerosol optical depth (AOD) and underlying liquid cloud optical depth, and is intended to be applied to measurements from sensors including the Sea-viewing Wide Field-of-view Sensor (SeaWiFS), Moderate Resolution Imaging Spectroradiometers (MODIS), and Visible Infrared Imaging Radiometer Suite (VIIRS). Together, these sensors provide around twenty years of observations to date. Airborne NASA Ames Spectrometers for Sky-Scanning, Sun-Tracking Atmospheric Research (4STAR) and NASA Langley High Spectral Resolution Lidar 2 (HSRL2) data collected during ORACLES provide important validation for spectral AOD for MODIS and VIIRS; as the SeaWiFS mission ended in 2010, it cannot be evaluated directly. These 4STAR and HSRL2 comparisons are complimentary and reveal performance generally in line with theoretical expectations. At present the two MODIS-based data records seem the most reliable, although there are differences between the deployments. Data collected in the region from other sources are also used to evaluate some assumptions made in the AAC retrieval. Spatiotemporal patterns in the data sets are very similar, and the time series themselves are very strongly correlated with each other (correlation coefficients from 0.95-0.99). Offsets between the time series are thought to be linked to differences in absolute calibration between the sensors, which can also explain some of the differences in validation results. The time series are also strongly correlated (correlations 0.78-0.94) with quantities such as ultraviolet aerosol index, total column AOD from



standard MODIS aerosol products, and active fire detections. This suggests that these quantities may also act as proxies for the above-cloud aerosol load in this region, when AAC retrievals are unavailable.

# 1 Introduction

Spaceborne monitoring of absorbing aerosols above clouds (AACs), typically smoke or mineral dust aerosols above liquid-phase clouds, has been a topic of increasing research interest in recent years. Yu and Zhang (2013) provide a review of the field, and Kacenelenbogen et al. (2018) a more recent list of approaches to their quantification. These AACs are important for multiple reasons. Their direct radiative effects can be very different from those above cloud-free surfaces (Hsu et al., 2003; Meyer et al., 2013; Zhang et al., 2014; Feng and Christopher, 2015), and they can have indirect and semi-direct effects on cloud formation, life cycle, and precipitation (Wilcox, 2012; Zhou et al., 2017). Their presence can lead to biases in retrieval of cloud optical depth (COD) and cloud effective radius (CER) if they are not accounted for, as they alter the brightness and spectral shape of the top of atmosphere (TOA) signal observed by passive sensors in a systematic way (Haywood et al., 2004). Additionally, they are largely missing from satellite aerosol optical depth (AOD) data sets derived from passive spaceborne imaging radiometers, which typically process only cloud-free scenes. Global aerosol and cloud fields tend to show similar regional and seasonal variations year after year, and AACs frequently occur downwind of some important aerosol source regions. These include, for example, smoke outflow from south-eastern Asia or southern Africa, as well as dust from the Sahara, Arabian Peninsula, and deserts in north-eastern Asia (e.g. Herman et al., 1997; Remer et al., 2008; King et al., 2013; Tsay et al., 2013; Lin et al., 2014). This interannual repeatability means that AOD data sets can have a persistent coverage gap in these regions, which biases estimates of the total atmospheric aerosol burden, and hinders aerosol transport analyses.

Semi-quantitative AAC observations from space began with the Total Ozone Monitoring Spectrometer (TOMS) sensor series, which used an ultraviolet aerosol index (UVAI) to take advantage of the spectral darkening of AACs (Herman et al., 1997). The large footprint size of TOMS (24-62 km at nadir dependent on sensor), however, was a limiting factor to quantitative applications. Similar observations are available from the Global Ozone Monitoring Instrument (GOME) sensor series. While simple to calculate, UVAI is only a semi-quantitative measure of AOD as it depends in a nonlinear way on factors including not only aerosol size, shape, altitude, and absorption, but also on the underlying surface (or cloud) properties and solar/view geometry (Hsu et al., 1999). Quantitative analysis benefited from the 2006 launch of the Cloud-Aerosol Lidar with Orthogonal Polarization (CALIOP), which is able to provide vertical profiles of aerosol/cloud backscatter and depolarization (Winker et al., 2013), and opened up a new era of quantitative spaceborne AAC research, processing algorithms (e.g. Chand et al., 2008; Costantino and Bréon, 2013; Meyer et al., 2013; Zhang et al., 2014; Alfaro-Contreras et al., 2016; Kar et al., 2018). More recently, this was supplemented by analyses based on the Cloud Aerosol Transport System (CATS) lidar on the international space station from 2015-2017 (Rajapakshe et al., 2017). While these sensors still have some limitations, the particular features





of AACs provide constraints which can obviate some of the assumptions required for these standard backscatter lidar aerosol retrieval algorithms (e.g. Hu et al., 2007; Kacenelenbogen et al., 2014, 2018; Liu et al., 2015), improving the quantification of AOD and lidar ratio for these cases.

Over the past decade or so, novel algorithmic techniques have been developed for spaceborne quantification of AACs from other sensors. These applications are often targeted to address one of the motivating factors listed above (Yu and Zhang, 2013). Torres et al. (2012) used the improved spatial, spectral, and radiometric capabilities of the Ozone Monitoring Instrument (OMI) over TOMS to use UVAI to make a more quantitative assessment of the AOD from AACs. This approach was subsequently refined using CALIOP data to improve regional assumptions about aerosol altitude, and clear-sky OMI data to improve assumptions about aerosol single-scattering albedo (SSA), enabling global application (Jethva et al., 2018). Jethva et al. (2013) also applied a conceptually-similar approach to Moderate Resolution Imaging Spectroradiometer (MODIS) measurements. de Graaf et al. (2012) used Scanning Imaging Absorption Spectrometer for Atmospheric Chartography (SCIAMACHY) data to estimate the radiative effect of smoke AACs above the south-eastern Atlantic. Here, AOD and COD were not directly retrieved, but rather the total shortwave radiative effect was estimated by considering separately those parts of the spectrum measured by SCIAMACHY strongly and weakly influenced by AACs, and inferring the aerosol-induced contribution. Meyer et al. (2015) developed an extension of the MODIS cloud optical properties retrieval algorithm for the south-eastern Atlantic, with a goal to remove the biases in retrieved COD and CER resulting from the lack of AACs in the standard MODIS cloud data set. Sayer et al. (2016) developed a similar technique but focused on filling AAC-related gaps in the Deep Blue (DB) aerosol data set. This was demonstrated with MODIS data, but was in principle also applicable to the Sea-viewing Wide-field of view Sensor (SeaWiFS) and Visible Infrared Imaging Radiometer Suite (VIIRS) sensors to which DB AOD retrieval algorithms have also been applied (e.g. Hsu et al., 2013). The Polarisation and Directionality of Earth's Reflectance (POLDER) instrument's multidirectional and polarimetric measurement capabilities provide greater information content for aerosols and clouds compared to single-view passive sensors. As a result, several POLDER-based techniques have also been used to quantify AACs (Waquet et al., 2013; Peers et al., 2015).

Much of this research has focussed on African biomass burning. From approximately June to October, agricultural fires move south from central Africa, generating large volumes of smoke which is transported into the south-eastern Atlantic Ocean where it passes over persistent large-scale marine stratocumulus cloud decks (Swap et al., 1996; Roberts et al., 2009; Zuidema et al., 2016). As an example, Figure 1 shows a case from September 4, 2017. Here, smoke (appearing greyish-brown) from widespread fires is seen blanketing much of Angola and northern Namibia, and covering part of a marine stratocumulus cloud deck which has formed along the coastline.

Taking a larger perspective, Figure 2 shows the long-term (2002 onwards) average daytime cloud fraction (from the MODIS Collection 6.1 cloud mask; Platnick et al., 2003), clear-sky total column AOD at 550 nm (from the MODIS Collection 6.1 Deep Blue/Dark Target merged product; Sayer et al., 2014b) and cloud-corrected overpass-corrected MODIS Collection 5 fire counts (Giglio et al., 2003, 2006) for the month of September. Intense burning across the continent causes large-scale AOD features over land, which are transported both over the stratocumulus deck to the west, and in a so-called 'river of smoke' to the southeast into the Indian Ocean (Swap et al., 2002, 2003; Kar et al., 2018). Discontinuities in the AOD field in this



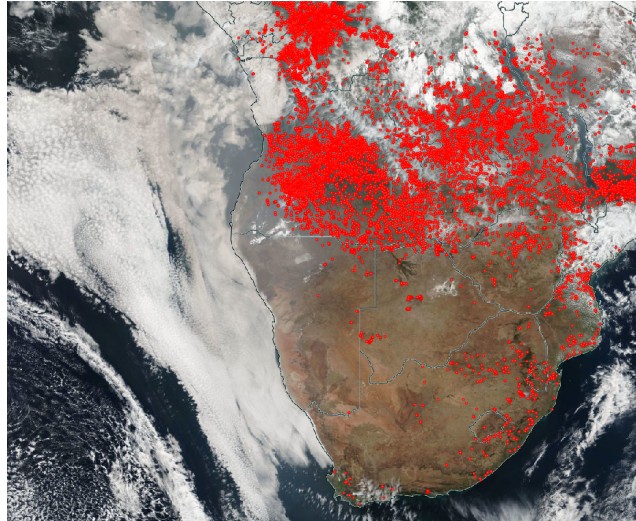

**Figure 1.** VIIRS true-colour image from September 4, 2017 showing smoke generated in central/southern Africa transported above marine stratocumulus clouds in the south-eastern Atlantic Ocean. Red dots indicate active fire detections. Region shown corresponds approximately to 36 °S-2 °S, 3 °W-38 °E. Image obtained from NASA Worldview, https://worldview.earthdata.nasa.gov.

composite are due in part to the different sampling of nearby land and ocean scenes, due to the coastal discontinuity in cloud cover, as well as land/ocean algorithm differences. Cloud fraction over portions of the Atlantic approaches 100 %, meaning few clear-sky AOD retrievals are possible; cloud cover over the southern Indian Ocean is lower.

These features make this region a natural laboratory for AAC studies, and several field campaigns have been carried out to
better understand aerosol-cloud-precipitation-radiation interactions in this region. Of most interest to the present analysis are the Southern African Regional Science Initiative (SAFARI) year 2000 campaign (Swap et al., 2002, 2003), and the ObseRvations of Aerosols above CLouds and their interactionS (ORACLES) campaign (Zuidema et al., 2016), which has deployments in different parts of the burning seasons from 2016-2018. These campaigns have included suites of airborne instrumentation for characterisation of these AACs, which have also provided invaluable data for the validation of AAC retrieval algorithms
(although this was not the primary purpose of these campaigns). Indeed, SAFARI-2000 data were used by Sayer et al. (2016) in the evaluation of the demonstration AAC retrieval algorithm further developed here. It is worth noting that additional field campaigns, with different foci related to the southern African aerosol/cloud system, have been carried out during the same period as ORACLES (Zuidema et al., 2016, 2018); these include Layered Atlantic Smoke Interactions with Clouds (LASIC), CLoud-Aerosol-Radiation Interactions and Forcing (CLARIFY), and AErosol RadiatiOn and CLOuds in Southern Africa
(AeroClo-SA). Deployments and flights generally took place within the area outlinted in green in Figure 2. The measurements from ORACLES are most directly suited to the evaluation of AAC retrieval algorithms, so are used here.

The purpose of this study is to describe updates to the initial AAC retrieval algorithm presented by Sayer et al. (2016), in preparation for its implementation in the DB aerosol data product suite, and use data collected during the 2016 and 2017



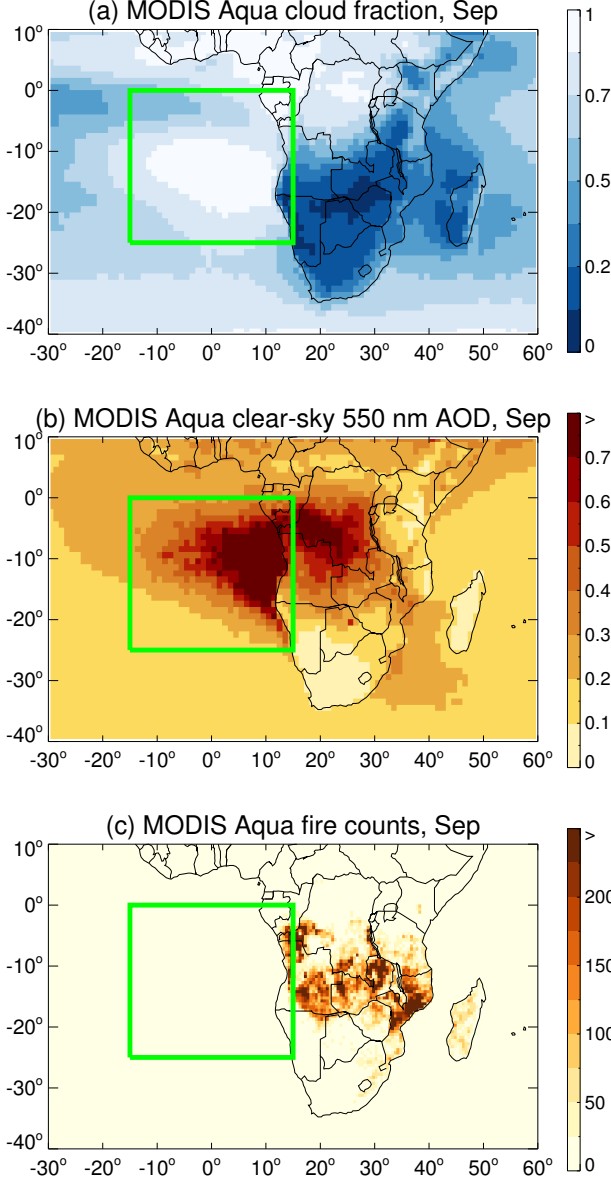

**Figure 2.** Long-term mean MODIS Aqua (a) daytime cloud fraction, (b) clear-sky total column AOD at 550 unitnm, and (c) total fire counts for the month of September for central and southern Africa and surrounding regions. Cloud and aerosol data are at 1° horizontal resolution, while fire counts are at 0.5° horizontal resolution. The green box (25 °S-0 °N, 15 °W-15 °E) denotes the approximate region of focus for the ORACLES campaign flights.

ORACLES deployments to further evaluate the algorithm. The study is organised as follows. Section 2 describes relevant features of the SeaWiFs, MODIS, and VIIRS satellite sensors, provides a summary of the retrieval algorithm introduced by




Sayer et al. (2016), and describes recent updates. Section 3 details the airborne data obtained during ORACLES and uses these observations to evaluate the updated AAC retrieval algorithm. Finally, the updated algorithm has been applied to process SeaWiFS, MODIS, and VIIRS observations across the large domain (40 °S-10 °N, 30 °W-60 °E) shown in Figure 2 from the start of the satellite missions to the end of 2017. Section 4 presents an initial look at this time series and compares results from

the different platforms. These AAC retrievals are available upon request to the authors. Note that a separate multi-algorithm comparison exercise is planned for a follow-up study; the purpose here is to introduce, evaluate, and examine the updated algorithm which will eventually be included within the DB data sets. The 2018 ORACLES deployment, and evaluation of the COD retrievals, will likewise be considered in a future study.

## 2    Satellite AAC retrieval algorithm summary and updates

### 2.1    Relevant sensor characteristics

Sayer et al. (2016) developed the AAC retrieval algorithm with a goal of implementation and error characteristics being as similar as feasible across the different sensors, relying on only those bands common to the three instrument types. SeaWiFS (McClain et al., 2004), MODIS (Barnes et al., 1998), and VIIRS (Cao et al., 2013) are all passive broad-swath imaging radiometers. SeaWiFS operated from late 1997 to December 2010; MODIS provides data on the Terra platform from late

February 2000, MODIS on the Aqua platform from July 2002, and VIIRS on the Suomi National Polar-orbiting Partnership (SNPP) from March 2012. Both MODIS sensors and SNPP VIIRS are still operational. SNPP VIIRS was followed with an additional VIIRS sensor launched in late 2017 (not considered in this study), and more are scheduled for the future.

SeaWiFS measured reflected solar radiation at the top of atmosphere (TOA) in eight bands with centres from 412-865 nm; MODIS and VIIRS have additional solar bands, as well as thermal infrared (tIR) channels. The AAC retrieval relies on common

bands centred near blue (470 nm for MODIS, 490 nm for SeaWiFS and VIIRS), green (550 nm), red (650 nm for MODIS, 670 nm for SeaWiFS and VIIRS) and near-infrared (nIR, 865 nm) wavelengths. These calibrated and geolocated measurements are referred to as Level 1b (L1b) data. Note that in this study these approximate wavelengths and/or band colour names (e.g. "green" for 550 nm) are sometimes referred to for simplicity, although all radiative transfer (RT) calculations use full sensor relative spectral response (RSR) functions. Specifically, the TOA reflectance $\rho$ for band $i$ is defined

$$\rho_i = \frac{\pi D_\odot^2 \int_0^\infty L_\lambda(\lambda) \Phi_i(\lambda) d\lambda}{\mu_0 \int_0^\infty E_\lambda(\lambda) \Phi_i(\lambda) d\lambda} \tag{1}$$

where $L_\lambda$ is the spectral radiance passing into the satellite field of view at TOA, $E_\lambda$ the downwelling solar spectral irradiance at TOA, and $\Phi_i$ the sensor RSR for band $i$, all functions of wavelength $\lambda$. The factor $D_\odot$ is the Earth-Sun distance in astronomical units (variable throughout the year), and $\mu_0$ the cosine of the solar zenith angle, which affect the total solar radiation received. Note that $L_\lambda$ and so $\rho_i$ depend on solar/observation geometry (and of course surface and atmospheric state), omitted here for

simplicity of notation.

Sensor pixel sizes are also similar. For MODIS, nominal horizontal pixel sizes vary from 0.25-1 km (dependent on band); here, the finer-resolution bands are aggregated and coregistered to the coarser bands at 1 km. For VIIRS, the nominal pixel size



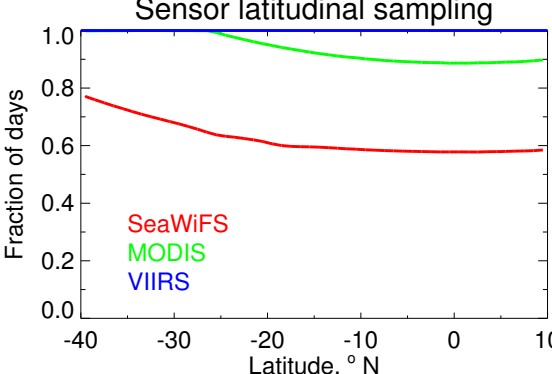

**Figure 3.** Fraction of days a given point (longitude) in the retrieval domain (Figure 2) is observed at least once by the individual SeaWiFS, MODIS, and VIIRS sensors, as a function of latitude.

for the relevant bands is 0.74 km. For SeaWiFS, pixel sizes are 1.1 km but on-board resampling performed for Global Area Coverage (GAC) mode subsamples these to provide an effective horizontal resolution of ∼4.4 km at nadir. As this GAC data are a subsampling rather than an average, it is most appropriate to consider these as 1.1 km pixels with gaps between them (as opposed to the continuous swaths of MODIS and VIIRS). All quoted pixel sizes are for nadir viewing geometries, at which

pixels are approximately square. Away from nadir the pixels enlarge, begin to overlap, and become more distorted in shape due to the scan geometry and Earth's curvature. This distortion and overlap is largest for MODIS (Sayer et al., 2015a), and smallest for VIIRS due to onboard mitigation strategies (Wolfe et al., 2013).

Swath widths are 1502 km for SeaWiFS (GAC mode), 2330 km for MODIS, and 3040 km for VIIRS (meaning VIIRS has no inter-orbit gaps). Around the Equator SeaWiFS also tilted to decrease the fraction of the swath affected by Sun glint in each

hemisphere; this tilt led to several scan lines near the Equator being missing. Figure 3 shows the resultant fraction of days when each of the sensors sampled a given location within the region, as a function of latitude. For SeaWiFS, coverage over the core of the stratocumulus deck (Figure 2) was obtained on about 60 % of days, potentially leading to larger sampling biases than the other sensors. For MODIS this value is closer to 85 % at the Equator and becomes 100 % poleward of ∼ 25°; for VIIRS, the whole region is imaged at least once per day. All the sensors are on platforms in Sun-synchronous polar orbits; MODIS

Terra has a daytime Equatorial crossing time of 10:30 (local solar time) while MODIS Aqua and VIIRS have an overpass time of approximately 13:30 local solar time. SeaWiFS had a crossing time near local noon at launch, although this slowly drifted in the later years of the mission (and ended around 13:30-14:00 in 2010).

The DB aerosol retrieval algorithm (Hsu et al., 2013) has also been applied to all these sensors to retrieve AOD for cloud-free scenes over land. The main data product from DB is the AOD at 550 nm; in this study mentions of AOD without a specific

wavelength indicated refer to 550 nm. For the SeaWiFS and VIIRS applications of DB (but not MODIS, at present), a Satellite Aerosol Retrieval Algorithm (SOAR) is applied over water surfaces to provide a near-global picture (Sayer et al., 2012, 2018). This combination of DB and SOAR is often colloquially referred to as the "Deep Blue" data product suite, even though DB





and SOAR are separate algorithms which use different bands and assumptions due to the differing characteristics of the aerosol retrieval problem over land and water surfaces.

This study uses the latest L1b data versions. For SeaWiFS this is obtained from the SeaWiFS operational Data Analysis System (SeaDAS) software package version 7.5 (available at https://seadas.gsfc.nasa.gov/). SeaDAS applies vicarious calibra-

tion coefficients obtained as described in Franz et al. (2007) to SeaWiFS TOA reflectances. For MODIS and VIIRS the current L1b data versions are Collection 6.1 (C6.1) and version 2 respectively. The main difference between C6.1 and the previous Collection 6 (C6) L1b data is the development and implementation of a fix for time-dependent crosstalk in MODIS Terra's thermal infrared (tIR) bands due to sensor degradation, which hindered the performance of the cloud mask in C6 in recent years (Moeller et al., 2017). Note that for the VIIRS application of SOAR, and for the AAC retrievals discussed here, VIIRS

TOA reflectances are cross-calibrated to bring them into radiometric consistency with MODIS Aqua using the method and coefficients of Sayer et al. (2017); the residual uncertainty on this correction is approximately 0.5-1 % for the bands used here (not counting any error on the MODIS Aqua calibration itself).

The DB/SOAR aerosol retrieval processing uses these L1b data at full resolution, but provides output level 2 (L2, geophysical data) products at coarser resolution. These L2 aggregations are 3×3, 10×10, and 6×6 L1b pixels for SeaWiFS, MODIS, and

VIIRS, respectively, giving L2 at-nadir horizontal pixel sizes of 13.5, 10, and 6 km respectively. To distinguish between native L1b pixels and the coarser resolution used in the L2 products, these latter sizes are often known as L2 "cells" rather than "pixels".

## 2.2    Summary of the Sayer et al. (2016) AAC retrieval algorithm

The physical principle behind the demonstration AAC retrieval algorithm presented in Sayer et al. (2016) is that, in the presence

of light-absorbing aerosols above a liquid-phase cloud, increases in COD brighten the TOA signal (as clouds tend to be bright and white) while AACs darken the signal as AOD increases. This darkening is more pronounced at the shorter wavelengths due to the tendency for increased absorption AOD (AAOD) at shorter wavelengths. For smoke aerosols this is due to the rapid increase of AOD with decreasing wavelength, while for dust it arises from the low SSA (strong absorption) at blue and green wavelengths but SSA close to 1 at red and nIR wavelengths. As a result, quantitative information about AOD and COD can be

extracted for AACs from the magnitude and spectral shape of TOA reflectance across this wavelength range (470-870 nm).

Sayer et al. (2016) retrieved AOD and COD at 550 nm (the state vector, $\mathbf{x}$) simultaneously by a weighted multispectral least-squares fit of TOA reflectances in the four (blue, green, red, nIR) bands to modelled TOA reflectances stored in a lookup table (LUT). The RT calculations used to create the LUT were performed with a tool based on the Vectorised Linear Discrete Ordinates (VLIDORT) code (Spurr, 2006). The same VLIDORT-based tool is used in the present study. The solution is found

by iterative minimisation of the squared residuals (differences between measured and LUT TOA reflectances) using the Optimal Estimation (OE) technique (Rodgers, 2000). OE propagates measurement and forward model uncertainties to provide an estimate $\mathbf{S}_{\mathrm{x}}$ of the uncertainty on the retrieved state $\mathbf{x}$,

$$\mathbf{S}_{\mathrm{x}} = \left( \mathbf{K}^{\mathrm{T}} \mathbf{S}_{\mathrm{y}} \mathbf{K} \right)^{-1},$$    (2)





where the covariance matrix $\mathbf{S}_y$ represents the uncertainty on the measurements (both radiometric and terms arising from forward model assumptions), and $\mathbf{K}$ (also known as the weighting function or Jacobian matrix) is the gradient of observations with respect to state measurements at the solution. More detail is given in Section 3.1 of Sayer et al. (2016). OE also provides a metric describing the level of agreement between measurement and modelled reflectances at the retrieval solution relative to

the expected level of disagreement (retrieval "cost", $J$), which is what is minimised iterativively in the retrieval:

$$J(\mathbf{x}) = \left(\mathbf{y}_m(\mathbf{x}) - \mathbf{y}_{LUT}(\mathbf{x})\right)^{\mathrm{T}} \mathbf{S}_y^{-1} \left(\mathbf{y}_m(\mathbf{x}) - \mathbf{y}_{LUT}(\mathbf{x})\right) \tag{3}$$

This is the sum of square residuals between measured ($\mathbf{y}_m$) and modeled LUT ($\mathbf{y}_{LUT}$) reflectances relative to their expected magnitudes given in $\mathbf{S}_y$, for a given point $\mathbf{x}$ in state space (i.e. combination of AOD and COD). Assuming $\mathbf{S}_y$ has realistic values and the measurements are informative on the state variables, $J$ is expected to take values around the number of degrees

of freedom (here two, as four measurements are being used to retrieve two parameters). These metrics are useful for quality assessment (QA) of the retrieval output; it is, however, important to note that they are only quantitatively robust if the underlying forward model is appropriate, the input uncertainties well-quantified, and the forward model approximately linear near the solution (Povey and Grainger, 2015). OE can optionally also account for *a priori* information on the state vector, but that has not been included in the present implementation. LUTs are interpolated linearly during the retrieval, and $\mathbf{K}$ calculated

numerically. The first guess at $\mathbf{x}$ is taken as the LUT node point with the lowest cost, and convergence is typically obtained within 3-4 iterations.

Based on typical features of aerosol/cloud systems, instrument capabilities, sensitivity analyses, and retrieval simulations, the RT forward model is set up as follows (cf. Section 3.1 of Sayer et al., 2016, and references therein). The cloud is assumed to consist of a homogeneous and fully overcast layer with a top altitude of 1.5 km above surface level, geometric thickness of

0.3 km, and be composed of liquid water droplets with size conforming to a gamma distribution with an effective radius of 12 μm and effective variance of 0.1. The underlying surface is assumed to be Lambertian (see also Sections 2.3.4 and 2.3.5). The aerosol is assumed to lie in a homogeneous layer with a top height 1 km above the cloud top, and be 0.5 km thick.

Sayer et al. (2016) considered six different optical models for AACs, corresponding to four different types of smoke aerosols from different source regions, and dust aerosols with two different SSAs. These optical models were based on results from

Aerosol Robotic Network (AERONET) almucantar scan inversions (Dubovik and King, 2000) thought representative of various source regions and aerosol types. Other sources of AACs such as volcanic ash may be important at certain locations and times, but were not included, due to their comparative rarity and the fact that they have less repeatable and well-defined optical properties. Retrieval simulations indicated that the information content of the measurements was such that it is not always possible for the retrieval to select the correct aerosol type out of the six using the cost function alone. Therefore in this study

the optical model representing strongly-absorbing smoke derived from AERONET inversions in Mongu (Zambia) is used in all cases. Based on previous studies of this region (Figure 2) this is expected to be reasonably representative of typical optical properties of the smoke AACs encountered in the study region (Piketh et al., 1999; Eck et al., 2003, 2013; Swap et al., 2003; Reid et al., 2005; Queface et al., 2011). An exception is periodic additional contributions from mineral dust in the northern





**Table 1.** Spectral complex refractive index for the smoke aerosol optical model used in this study, after Sayer et al. (2014a).

| Wavelength | Fine mode | Coarse mode |
|---|---|---|
| 440 nm | 1.51-0.024i | 1.45-0.0035i |
| 675 nm | 1.52-0.022i | 1.45-0.0015i |
| 870 nm | 1.52-0.021i | 1.45-0.0015i |
| 1020 nm | 1.52-0.021i | 1.45-0.0015i |

part from December-February (Pandithurai et al., 2001; Ben-Ami et al., 2009). Over the main ORACLES domain (green box in Figure 2), and during the peak burning season, however, other AAC sources are expected to be negligible.

The Mongu smoke aerosol model was described by Sayer et al. (2014a). It is a bimodal lognormal optical model, such that fine (subscripted f throughout) and coarse (subscripted c throughout) volume size distributions are each of the form

$$\frac{dV(r)}{d\ln(r)} = \frac{C_{\mathrm{v}}}{\sqrt{2\pi}\sigma} e^{-\frac{1}{2}\left(\frac{\ln(r) - \ln(r_{\mathrm{v}})}{\sigma}\right)^2}, \tag{4}$$

for particles of size $r$, given total mode particle volume $C_{\mathrm{v}}$, mode (which is also median and geometric mean) radius $r_{\mathrm{v}}$, and width $\sigma$. Sayer et al. (2014a) found that the modal radius (in $\mu$m) of the fine mode was dependent on the fine-mode AOD at 550 nm ($\tau_{\mathrm{f}}$) as follows:

$$r_{\mathrm{v,f}} = 0.161 + 0.013 \ln(0.63\tau_{\mathrm{f}}) \tag{5}$$

The spread $\sigma$ (dimensionless) of the fine mode was also found to have a weak dependence on ($\tau_{\mathrm{f}}$),

$$\sigma_{\mathrm{f}} = 0.469 + 0.023 \ln(0.074\tau_{\mathrm{f}}), \tag{6}$$

i.e. for higher smoke loadings the fine-mode particles were larger on average and had a broader variety of sizes. In contrast, $r_{\mathrm{v,c}}$ and $\sigma_{\mathrm{c}}$ were found to be AOD-independent (across the small range of coarse-mode AOD observed) and take typical values of 3.34 $\mu$m and 0.67 respectively. Further, Sayer et al. (2016) assumed a representative fine mode fraction (FMF) of AOD at 550 nm, i.e. $\tau_{\mathrm{f}}/(\tau_{\mathrm{f}} + \tau_{\mathrm{c}})$, of 0.9 for these smoke AACs, based on typical values from Sayer et al. (2014a).

The assumed aerosol refractive index is shown for AERONET wavelengths in Table 1. These values are interpolated in log-log space to the satellite band centres for calculation of aerosol phase matrix elements and SSA. The resulting overall SSA is weakly dependent on the AOD, but in general varies from approximately 0.86 in the blue band to 0.82 in the nIR band. This is discussed in more detail later.

## 2.3 Algorithm updates since Sayer et al. (2016)

The same overall retrieval approach and RT forward model described in Section 2.2 is used in the present study, with some updates as described below. These updates are intended to better account for approximations and assumptions made in Sayer





et al. (2016), in particular to retrievals for clouds above land surfaces, and to prepare the AAC retrieval for integration with the standard DB and SOAR AOD retrieval data products.

### 2.3.1 Pixel selection and aggregation

For the two test case scenes in Sayer et al. (2016), the AAC retrieval algorithm was applied to MODIS data at full (nominal

1 km) resolution. Here, to prepare for integration into the main DB/SOAR AOD data sets, the retrieval is instead performed at the equivalent pixel aggregations for the L2 products for each sensor (cf. Section 2.1). This is achieved by taking the median spectral TOA reflectance for suitable L1b pixels within each L2 cell. Use of medians rather than means decreases sensitivity to e.g. cloud masking errors and 3D RT effects which are not accounted for by the forward model (Várnai and Marshak, 2002; Cho et al., 2015). A cell is only processed if the proportion of suitable pixels within the cell is greater than 75% (i.e. at least

75/100 for MODIS, 48/64 for VIIRS, or 7/9 for SeaWiFS), as the forward model is expected to be less appropriate for broken clouds.

For the averaging of TOA reflectance, a suitable pixel is defined as one which is thought to represent a liquid-phase cloud (either with or without an overlying absorbing aerosol layer). For MODIS Terra and Aqua, the standard cloud mask product is used, and cloud phase is taken from the standard MODIS cloud optical properties data sets (Platnick et al., 2003; Frey

et al., 2008). Both of these are available within the C6.1 MOD06_L2 (for Terra) and MOD06_L2 (for Aqua) data files. For VIIRS, the equivalent cloud mask product (VNPCLDMK) is used from the current version 1. No VIIRS cloud phase product is available at the time of writing, so water clouds were identified empirically by assuming that any cloudy pixel with a brightness temperature (BT) in the VIIRS 12 μm band below 270 K corresponded to an ice or mixed-phase cloud and discarding it. While empirical, this threshold seems appropriate in this case based on manual examination of the data, as the vast majority of AAC

cases in this region correspond to marine stratocumulus clouds with warmer BTs. For both MODIS and VIIRS, only pixels identified as "probably cloudy" or "confidently cloudy" are considered.

SeaWiFS has no equivalent cloud mask product, and lacked the tIR bands useful for determining cloud phase. As a result, a separate cloud mask has been developed, drawing from that developed for the DB/SOAR SeaWiFS aerosol products (Hsu et al., 2004, 2013; Sayer et al., 2012). Cloud masking for AOD retrievals is generally designed to be conservative in that it minimises

the chance of the retrievals being contaminated by an undetected cloudy pixel. Here, the focus is different, as the desire is to retain optically-thick clouds which are likely to be liquid water, and so tests and thresholds are modified, although follow similar principles to the above references. Specifically, land and water surfaces have different TOA reflectance brightness tests, such that a pixel is marked as cloudy and suitable if

$$\rho_{412} \geq \frac{0.1\mu_0}{\pi} \tag{7}$$

over land, or

$$\rho_{865} \geq \frac{0.07\mu_0}{\pi} \tag{8}$$

over water. The factor of $\mu_0$ in the numerator accounts for the fact that reflectance approaches infinity as the Sun approaches the horizon (Equation 1), while with this normalisation the reflectance of an optically-thick cloud is less dependent on solar





zenith angle. The specific bands chosen for land and water are those at which the surface reflectance tends to be smallest, offering the best discrimination between cloudy and cloud-free scenes, and thresholds robust to the presence of AACs.

These tests have been found to be fairly effective at identifying optically-thick clouds. As there is no cloud phase mask for SeaWiFS, additional tests are implemented to identify optically-thin clouds (such as cirrus, but also residual optically-thin
liquid phase). This is based on spatial variability at the 412 nm band (where clouds tend to show greater spatial variability than cloud-free scenes). This considers 3×3 groupings of L1b pixels, marking the central pixel cloudy if the test is passed, and again has separate tests for land and water pixels. Over land, the pixel is marked as cloudy but unsuitable if the ratio between the maximum and minimum reflectance at 412 nm is greater than 1.5, but the absolute brightness test is not passed. Over water, it is marked as cloudy but unsuitable if the standard deviation of reflectance over water pixels within the 3×3 pixel box is greater
than $0.01\mu_0/\pi$, but the absolute brightness test is not passed. Again, these tests have heritage in the SeaWiFS DB (Hsu et al., 2004, 2013) and SOAR (Sayer et al., 2012) data sets.

Only detected clouds passing the brightness tests are processed with the AAC retrieval algorithm for SeaWiFS (provided the cell they are in meets the 75 % suitability threshold described above). The TOA reflectance thresholds remove optically-thin cirrus clouds from consideration, and output QA filtering (described below) removes others. The spatial variability tests are
intended to provide a summary view of the true (i.e. total suitable plus unsuitable) cloud fraction, for comparison with the other sensors. However, this limitation does mean that the total cloud fraction and suitable pixels on which AAC retrieval is attempted may differ between SeaWiFS and the MODIS/VIIRS applications of the algorithm.

### 2.3.2 Surface elevation

In Sayer et al. (2016), the two MODIS test cases examined were predominantly over water, for which the assumption of 1
standard atmosphere pressure is reasonable. This is not necessarily the case over land; Figure S1 shows that much of the study region is above 1 km in altitude. Not accounting for this has the potential for regional biases in the algorithm results, as atmospheric pressure determines the total Rayleigh scattering and its interaction with atmospheric multiple scattering and absorption. This could be particularly evident across land/ocean boundaries, e.g. off the coasts of Namibia and Angola where the stratocumulus deck is often encountered.

As a result, an additional dimension has been added to the retrieval LUT to account for elevation-dependent changes in surface pressure. Surface elevation ($z$) provided within the L1b files for each pixel is converted to surface pressure ($p$) according to the relationship

$$p = p_0 e^{\left(\frac{-z}{H}\right)}, \tag{9}$$

where the reference pressure $p_0$ is taken as 1013.25 mbar and the atmospheric scale height $H$ is assumed to be 7.4 km (sen-
sitivity to this number is small). The LUT contains nodes at 1013.25, 700, and 400 mbar surface pressure, sufficient to cover the range of elevations encountered here with minimal (generally <0.5 %) interpolation error in TOA reflectance, and is (as in other dimensions) interpolated linearly.



### 2.3.3 Ancillary meteorological data

As in the routinely-produced DB/SOAR AOD data sets, ancillary meterological data are needed to correct the TOA reflectance for absorption by trace gases (for the bands considered here, chiefly $H_2O$ and $O_3$) and provide a near-surface (10 m) wind speed to calculate Sun glint reflectance over water (see Section 2.3.5). For MODIS and VIIRS, these are obtained from the NASA
Goddard Earth Observing System Model, Version 5 (GEOS-5, Rienecker et al., 2008) Forward Processing for Instrument Teams (FP-IT) data stream, available from http://gmao.gsfc.nasa.gov/products, which is also used in VIIRS DB/SOAR data processing (Sayer et al., 2018). This begins in 2000, so is unavailable for the inital years of the SeaWiFS mission; as a result, the Modern-Era Retrospective analysis for Research and Applications, Version 2 (MERRA2, Gelaro et al., 2017), available from https://gmao.gsfc.nasa.gov/reanalysis/MERRA-2, is used for the full SeaWiFS record instead. MERRA2 is built on an earlier
version of the GEOS-5 model; for the quantities used here (column $H_2O$, column $O_3$, and 10 m) wind speed) the differences between FP-IT and MERRA2 are generally small and the differences introduce negligible additional uncertainty. These fields are at 0.5° latitude by 0.625° longitude resolution, with timesteps of 1 (MERRA2) and 3 (FP-IT) hours, and the data are interpolated linearly in space and time to each L1b pixel.

Trace gas absorption correction follows the method and coefficients of Patadia et al. (2018), as in VIIRS DB/SOAR data, for
MODIS and VIIRS. For SeaWiFS, coefficients from the SeaDAS software (which follows the same basic approach) are used. The purpose of this correction is to simplify retrieval LUT generation by removing the need to include variations in these gas absorbers within the LUTs. The assumption is made that trace gas absorption can be decoupled from other (Rayleigh, aerosol, cloud, surface, and their interaction) scattering and absorption. Then, the TOA reflectances are brightened by dividing by the estimated transmittance as a result of these absorbers, giving the TOA reflectance which would be observed in the absence of
these species. For $O_3$ this is reasonable because the bulk of the absorption occurs in the stratosphere and is separated from the bulk of the atmospheric signal; in addition, ozone varies fairly smoothly in space and time. For $H_2O$ this is less valid as water vapour varies on finer spatiotemporal scales, and is more heterogeneous in its vertical distribution through the atmosphere. Here, as in Sayer et al. (2017, 2018), the assumption is made that half the water vapour lies below the cloud (and is not seen) and half above. For the bands used in the AAC retrieval, $H_2O$ absorption is fairly weak and, except for the nIR band, $O_3$ is the
dominant absorber. Total atmospheric gaseous transmittance varies dependent on band, solar/view geometry, and atmospheric constituents, but generally ranges from ∼0.99 (for the blue bands) to ∼0.8 (for a low Sun and oblique view in the green bands, with a high ozone concentration). Hence, while large errors in the $O_3$ absorption correction are thought to be unlikely, a larger potential error of order 50 % in $H_2O$ absorption is still expected to cause an error of only 1 % or less in TOA reflectance at these bands.

### 2.3.4 Land surface reflectance

When a cloud is opaque, the TOA reflectance across the visible part of the spectrum is largely insensitive to the underlying surface albedo. Hence, the demonstration algorithm in Sayer et al. (2016) made the simplifying assumption of a spectrally-neutral surface albedo of 0.05 in all bands. However, when the cloud is optically thin, there is a surface contribution to the TOA





signal and assumptions about surface albedo become more important. While the QA tests described below filter out low-COD scenes, if the underlying surface reflectance is brighter than assumed, it is possible that a low-COD cloud could be erroneously retrieved as a combination of higher-COD with a higher AOD and pass the QA tests under some circumstances. As a result, in the present study, the surface albedo assumption over land is updated using a climatology derived from MODIS data.

For this purpose the gap-filled snow-free albedo product (MCD43GF, Sun et al., 2017) is used as a basis. MCD43GF is derived using the MODIS bidirectional reflectance distribution function (BRDF) Terra/Aqua combined product (MCD43A1, Schaaf et al., 2002) and applying additional filtering and spatial/temporal constraints to provide BRDF model parameters at 30 arc second resolution, once per eight days from 2003-2015. Note that the inputs used for the currently-available version of MCD43GF derive from the MODIS Collection 5 processing.

While MCD43GF provides full BRDF model parameters, for computational simplicity these are used to calculate white-sky (Lambertian) albedo for use in the retrieval forward model. This approximation is justifiable because under a cloudy sky it is likely that most of the light field below the cloud will be diffuse. For example, for a COD of 1 and vertical incidence only 37 % ($\sim e^{-1}$) of photons entering the top of the cloud will be directly transmitted without being scattered at least once or absorbed. Above- and below-cloud aerosol and Rayleigh scattering and absorption will further decrease this proportion.

Additionally, to decrease the storage overhead and enable processing outside the 2003-2015 time frame, the source MCD43GF are spatiotemporally aggregated to provide a data base for a representative year (retaining the 8-day time steps) at 0.05° resolution. The spatial aggregation is done first, taking the source MCD43GF products and recording the median albedo within each 0.05° grid cell. After the spatial aggregation, for each grid cell, spectral band, and eight-day period (out of 46 in a year), the median albedo from up to 13 years is taken as representative of that location and time of year. This collapses the interannual 20    variation to provide, for each point, the annual cycle of surface albedo, which is used in the AAC retrieval.

As a measure of the uncertainty introduced by the spatial coarsening, Figure S2 shows the mean of the spatial standard deviation of albedo within each grid cell. For all bands except 865 nm, this is generally small (<0.02); even at 865 nm, generally <0.04. This indicates reasonable homogeneity of surface brightness on these scales. The exceptions tend to be salt pans, e.g. the Makgadikgadi Pans in Botswana. Figure S3 shows the mean (across all eight-day periods) temporal standard 25    deviation (across the 13 years) of surface albedo, i.e. a measure of the interannual variability at each location. Spatial patterns are broadly in line with S2, although the magnitudes tend to be slightly higher. This is expected as interannual variability in weather patterns influences vegetation growth and harvest, which influences the surface albedo, especially at 865 nm, which is strongly linked to vegetation cover (Tucker, 1979). Sun et al. (2017) assessed the gap-filling procedure in MCD43GF by randomly removing input data and comparing the gap-filled result with that withheld data. For white-sky albedo, this gave root 30    mean square errors (RMSEs) of 0.020 and 0.027 for red and nIR bands respectively. These are similar to or smaller than the quadrature sum of the spatial and temporal aggregation variabilities shown in Figures S2 and S3. Many of the areas with higher spatiotemporal variability are also associated with lower cloud cover (e.g. Figure 2), meaning they are areas less likely to have an AAC retrieval in the first place, although it is possible that these regions do not represent real cases of smaller variability, but rather more cloudiness means less source data available as input to MCD43GF. Sun et al. (2017) did not show results for 35    blue or green bands, but based on the results here it is likely they would be similar or smaller. It is therefore reasonable to





assume that the method applied here to generate a climatological data base for the AAC retrieval does not significantly degrade the utility of the MODIS albedo product for this application. The resulting annual cycles of surface albedo are shown for four sample locations, representing different surface types, in Figure S4. In all cases the annual cycle tends to be larger than the interannual variability, which is encouraging as the year-to-year changes are neglected in the present approach.

In the AAC retrieval, surface albedo is assigned at full L1b resolution using the nearest 0.05° grid cell in the climatology from the 8-day period of the year into which the granule falls. Analogously to the aggregation of TOA reflectance, the cell median surface albedo is also used during the retrieval process. The same data base is used for all three sensors, as the source BRDF products are at present only available for MODIS, although an equivalent VIIRS data processing suite is in development. Differences in band centres and widths thus have the potential to introduce additional error for SeaWiFS and VIIRS retrievals

using this MODIS-derived data base, although these are expected to be smaller than 0.02. This is a second-order contribution to the total forward model error in terms of TOA reflectance, especially for optically-thick clouds.

### 2.3.5   Water surface reflectance

Analogously to the over-land surface reflectance treatment, the assumption of a spectrally-neutral albedo of 0.05 from Sayer et al. (2016) is also updated over water surfaces. The reflectance is instead modelled as a combination of a wind-roughened

surface using the wind-isotropic model of Cox and Munk (1954a, b), with the ancillary data described in Section 2.3.3 as input, added to a reflectance of 0.05, 0.04, 0.03, and 0.03 to represent ocean colour and whitecap contributions for the blue, green, red, and nIR bands respectively. Real deviations from this are expected to be of the order ±0.01, which is again a second-order contribution to the total forward model error in terms of TOA reflectance under optically-thin clouds, and becomes negligible for opaque clouds.

### 2.3.6   Retrieval QA

As in Sayer et al. (2016), QA metrics are used to filter the retrievals to remove scenes where the retrieval was not able to find a good fit between measured and modelled reflectances, or where unphysical spatial structure suggests that the forward model may have been inappropriate. These tests are similar to those described in Sayer et al. (2016), with some updates based on examination of the larger data volume processed for this study. Specifically, pixels are only retained if the following criteria

are all met:

- The retrieval cost function value (Equation 3) is less than 5, indicating that the forward model is able to match the spectral TOA reflectance reasonably well. In practice cost function values tend to cluster in the range 0-2 or else be much higher than 5, so the results are only weakly sensitive to the value of this threshold.

- The COD is 2 or higher, as for optically-thinner clouds the retrieval solution was found to often be ambiguous, and

more sensitive to errors in surface reflectance assumptions. These factors do not always lead to a high value of the cost function. This is a slight relaxation of the COD threshold of 4 used in Sayer et al. (2016), due to the improved surface reflectance models used in this work. It can increase the potential data volume by 50 % or more in some cases, although



some of these retrievals are subsequently removed by other QA tests, and the change in the threshold is also dependent on the larger-scale structure of the cloud field. Thus it is somewhat difficult to quantify the overall effect on data volume.

- The retrieval has two or more (out of a possible eight) neigbours. Based on manual inspection, cases with zero or one neighbours are often found in conditions of broken cloudiness (e.g. cloud fragments in the middle of open-celled stratocumulus), which again may mean the forward model is not appropriate but does not always result in a high retrieval cost.

- The absolute difference between the retrieved AAC AOD and the median of AOD retrieved in the the $3 \times 3$ retrieval box around it is smaller then 0.2. This removes isolated spikes of high- or low- AOD which can result from isolated thin clouds, cloud mask errors, or poor surface assumptions. In practice, these retrievals are often around the edges of cloud fields. The physical basis behind this is that the AOD fields are expected to be spatially smooth on the scales of several retrievals. Note that the OE-provided uncertainty estimates (Equation 2) for these retrievals are often (but not always) large. Sayer et al. (2016) implemented a test based instead on the estimated retrieval relative uncertainty, which had similar results for high-AOD artefacts, but was less effective at identifying low-AOD outliers. This test might be less appropriate in other regions of the world where spatial variability in the aerosol field is higher.

An example showing the overall QA flag and results for individual tests is given in Figure 4. This granule was one of the test cases compared by Sayer et al. (2016) against airborne data in SAFARI-2000; in that case, the airborne measurements gave an estimate of the above-cloud AOD at 550 nm of 0.49, with a spatiotemporal standard deviation of 0.04, within the area outlined by a green box in Figure 4(a). The current version of the algorithm retrieves mean (median) AOD across this box of 0.48 (0.51), in very good agreement, and negligibly different from the results of the demonstration algorithm shown in Sayer et al. (2016). The small difference from those prior results for this example is expected as the only relevant differences are the updates to the MODIS L1b data version (Collection 6 to 6.1) and aggregation/QA tests. Sayer et al. (2016) noted that the good agreement (AOD within 0.02) for this case may be fortuitious as the estimated uncertainty on the retrieved AOD (cf. Equation 2) is ±0.18, somewhat larger.

## 3  Validation with airborne data during ORACLES

### 3.1  Instrumentation

#### 3.1.1  NASA Ames Spectrometers for Sky-Scanning, Sun-Tracking Atmospheric Research

The NASA Ames Spectrometers for Sky-Scanning, Sun-Tracking Atmospheric Research (4STAR) instrument is an aircraft-mountable hyperspectral sunphotometer and sky radiometer (Dunagan et al., 2013). It is a successor to the multispectral Ames Airborne Tracking Sunphotometer (AATS) instruments (for some AATS field campaign deployments see e.g. Redemann et al., 2003; Schmid et al., 2003), which were used by Sayer et al. (2016) in validation of the initial version of the Deep Blue AAC retrieval algorithm (and cf. Figure 4). 4STAR combines the sun-tracking ability of AATS with a sky-scanning ability similar



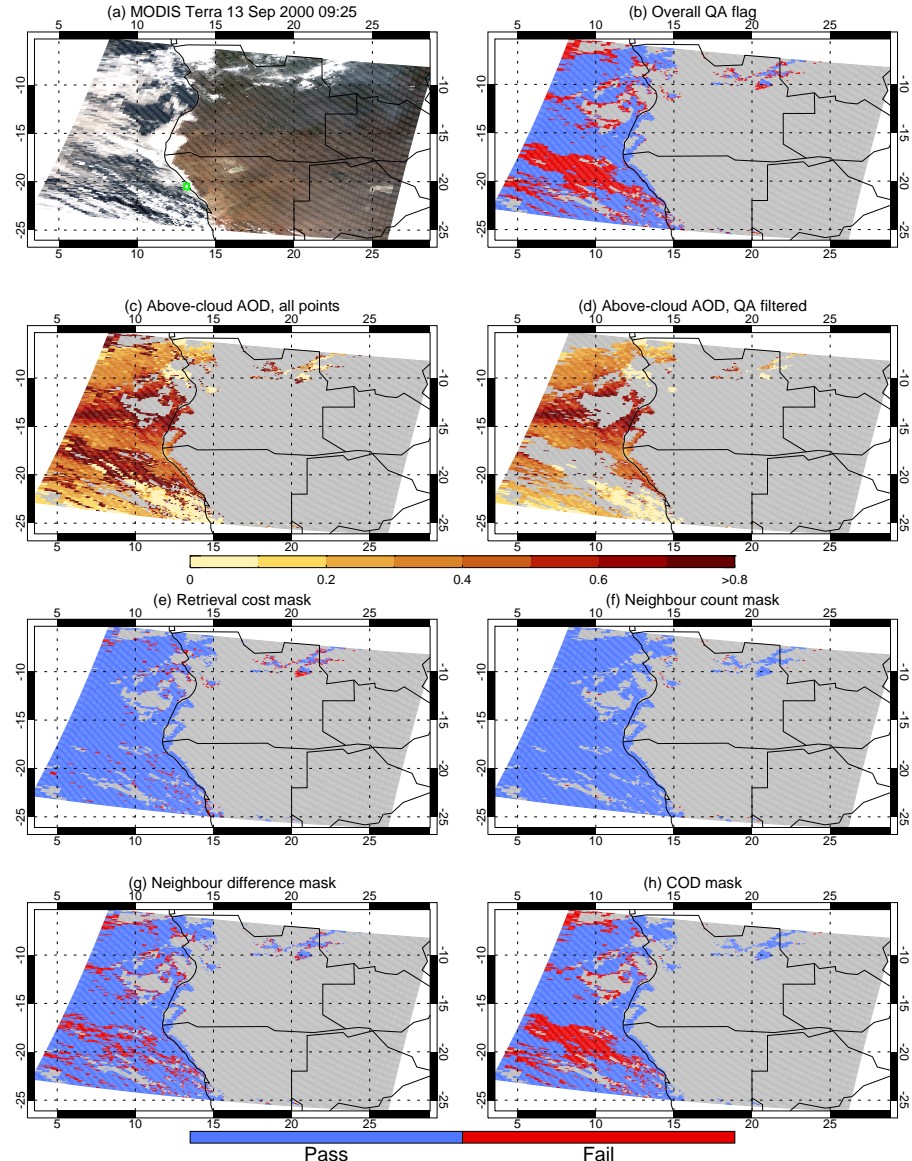

**Figure 4.** AAC retrieval for a MODIS Terra granule during SAFARI-2000 from September 13, 2000. Panels show (a) a true-colour image, (b) the overall QA flag, (c, d) the retrieved AOD before and after applying QA tests, and (e-h) results of individual QA tests, as described in the text. The colour classification for panel (b) is the same as (e-h). The green box in (a) shows the region used for comparison with airborne data by Sayer et al. (2016). Pixels without valid retrievals are shaded in grey.

to that of the ground-based AERONET sun/sky photometers. Its full-width field-of-view (FOV) when measuring direct solar beam irradiance is 2.4° (LeBlanc et al., 2019), with a radiometric deviation of less than 1 % in this span, while AATS showcases a slightly larger full-width field-of-view of 3.7° (Segal-Rozenhaimer et al., 2013). The smaller FOV reduces uncertainties due





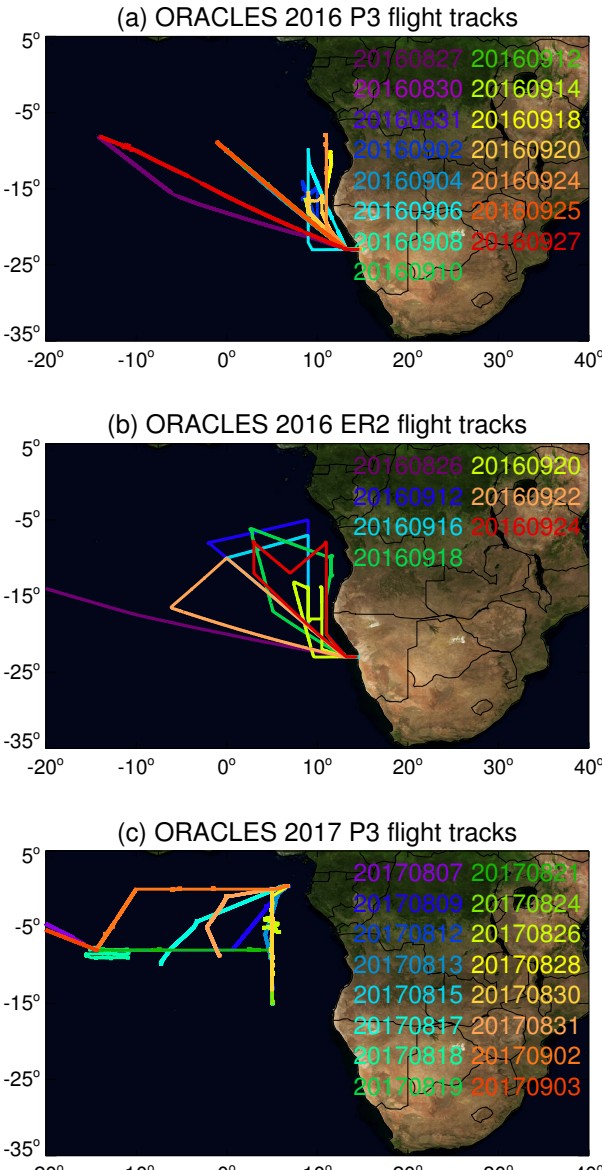

**Figure 5.** Flight tracks for the 2016 and 2017 ORACLES deployment, coloured by date. From top to bottom, panels indicate the 2016 P3, 2016 ER2, and 2017 P3 aircraft flight tracks.

to scattered light in the direct-beam signal (Segal-Rozenhaimer et al., 2013; Smirnov et al., 2018). 4STAR's hyperspectral measurements have 1556 overlapping and continuous bands ranging from 350-1700 nm, compared to 6 or 14 distinct non-overlapping spectral bands on the AATS instruments (Dunagan et al., 2013).





The instrument was mounted on the NASA P3 aircraft for both the 2016 (based out of Walvis Bay, Namibia) and 2017 (based out of Sao Tomé) ORACLES deployments. Flight tracks where scientific data were collected are shown in Figure 5. More information about the 2016 deployment, as well as 4STAR-derived aerosol data, can be found in LeBlanc et al. (2019). Flights included spiral profiles through smoke layers above clouds, to enable the airborne instrumentation to measure atmospheric properties at different points within the smoke layers. The data set includes a flag (described in LeBlanc et al.,

2019) to indicate measurements when the aircraft was flying just above the top of a cloud and below a smoke layer (typically at the bottom of one of these spirals). These flagged data points comprise a fairly small fraction of the total data set, but allow estimates of the above AOD suitable for validation of the satellite retrievals.

The 4STAR data product used here is the spectral AOD from direct-sun measurements. Exact measurement characteristics can change between deployments, but in general the data are provided at around two dozen wavelengths (outside of strong gas

absorption features) with a temporal resolution of 1 s. The uncertainty in spectral AOD is estimated on a point-by-point basis, and is largely driven by uncertainties on radiometric calibration and trace gas absorption, but is typically of order 0.005-0.02 (decreasing with increasing wavelength). As this is much smaller than the expected uncertainty on the satellite retrieval, it is suitable as a validation data source. The current data versions used here are R3 for 2016, and R1 for 2017. Note that 4STAR measurements can also provide AERONET-like aerosol inversions and transmission-based cloud property retrievals (LeBlanc

et al., 2015) which will be considered in a separate study.

### 3.1.2    NASA Langley High Spectral Resolution Lidar

The second airborne instrument used is the NASA Langley High Spectral Resolution Lidar version 2 (HSRL2; Hair et al., 2008; Burton et al., 2018). HSRL2 provides vertical profiles of atmospheric backscatter, depolarisation, and extinction; an advance of the 2016 deployment was the addition of a 355 nm channel (Burton et al., 2018) alongside the 532 and 1064 nm channels.

Note that the 1064 nm channel lacks HSRL capability and is backscatter-only, so above-cloud AOD is only provided at 355 and 532 nm. An advantage of the HSRL technique is that it is able to determine both backscatter and extinction, removing the need for a lidar ratio assumption, which can be a large source of uncertainty in backscatter lidar AOD retrievals such as from CALIOP (Omar et al., 2009).

During the 2016 ORACLES deployment (Burton et al., 2018), HSRL2 flew on the NASA ER2 high-altitude aircraft (Figure

5), also based out of Walvis Bay, Namibia. The ER2 typically flew at an altitude around 20 km, well above the clouds and the bulk of the aerosols. As the ER2 was flying at high altitude, a larger proportion of the flight provide data suitable for validating the AAC algorithm compared to 4STAR, for which only data collected immediately above a cloud top are relevant. During the 2017 ORACLES deployment, the HSRL2 instrument flew on the P3 with 4STAR, at lower altitudes; this means that an unknown amount of aerosol above the plane will have been missed in the 2017 deployment. This should be borne in mind

when examining the 2017 matchup statistics. To decrease the likely contribution from this unobserved aerosol, 2017 HSRL2 data are only used here when the P3 was flying above 5 km (flight altitudes were generally below 8 km; when not spiraling, a reasonable number of legs were between 5-6 km, and few were above 6 km). The current data versions used here are R7 for 2016, and R1 for 2017.



Profiles are measured at 15 m vertical resolution and 10 s temporal resolution; the data contain a flag to identify profiles containing AAC cases. The spectral AOD was determined as described in by Hair et al. (2008), from the difference of the molecular channel signals at the top of the profile and at the top of the cloud. Assessment of the uncertainties of AOD determined from HSRL2 data are provided by Hair et al. (2008) and Burton et al. (2018). In brief, there is a random component (which is quantified within the data and typically negligible, $\ll 0.01$) and a larger, locally-systematic component. This sys-

tematic component is expected to be dominated by uncertainties in the molecular profile used in the retrieval, and is difficult to quantify. As such, the uncertainty in HSRL AOD data are typically estimated by comparing against other simultaneous observations. Rogers et al. (2009) evaluated 532 nm AOD from an older version of the HSRL instrument in Mexico City and found a RMS difference of 0.008 against AATS data and 0.056 against AERONET; in the latter case, some of the disagreement with AERONET was thought to result from small amounts of aerosol above the plane's flight altitude, and one outlying point

(of only 10 total) contributed disproportionately to the higher RMS difference. Sawamura et al. (2017) found a similar level of agreement against AERONET from HSRL-2 at both 355 and 532 nm from two field campaigns over urban areas in California and Texas.

## 3.2 Validation approach

Only retrievals passing the QA tests described in Section 2.3.6 are considered. As the airborne data have a higher spatial and

temporal resolution than the satellite retrievals, the satellite data are validated by checking for and aggregating the 4STAR and HSRL2 data inside each individual retrieval footprint. Although this leads to a large number of matchups, it is important to bear in mind that the resulting matched data are not independent, due to the large autocorrelation in the underlying aerosol field, and retrieval errors are similarly likely to be autocorrelated on these length scales. The airborne data are available only for a limited spatial domain over a short time period within each year. This is a different picture from total column AOD validation

using ground-based AERONET sites, which are composed of individual dispersed sites as opposed to flight tracks. For this reason, as well as statistics for all matchups individually, granule-average statistics (i.e. statistics calculated using averages of all matchups from individual granules) are also presented. These should exhibit reduced autocorrelation compared to the all-matchups data. Note that these are calculated averaging all matched retrievals and airborne data within individual granules, not simply averaging all retrievals within the granules.

In many cases the satellite overpasses and flight tracks were not simultaneous, and a time difference threshold of $\pm 3$ hours is used as a cutoff for a matchup to be valid. This is somewhat longer than the $\pm 0.5$-1 hours often used for comparison against AERONET sites, and is adopted here as the temporal variability of these large-scale smoke plumes is expected to be somewhat limited. Part of the rationale for a shorter time threshold in AERONET validation analyses is the potential for an incoming cloud field to remove or modify the aerosol during the time between measurement and overpass; as the AAC retrieval is by

nature concerned with those aerosols above (and less likely to be modified by) clouds, that rationale is less relevant here. Using a stricter time threshold in this analysis essentially has the effect of removing individual flight legs from consideration; due to the limited number of flights available (Figure 5), it is difficult to disentangle contributions from true temporal variability from those due to individual flight characteristics (i.e. sampling differences) in the changes in resulting comparative statistics,





**Table 2.** Number of individual retrieval matchups (and number of contributing granules, in parentheses) for each satellite sensor and ORACLES data set.

| ORACLES | Count | | |
|---|---|---|---|
| data set | MODIS Terra | MODIS Aqua | VIIRS |
| 4STAR, 2016 | 532 (20) | 432 (15) | 835 (17) |
| 4STAR, 2017 | 190 (12) | 285 (15) | 561 (18) |
| HSRL2, 2016 | 1918 (13) | 1896 (14) | 4441 (14) |
| HSRL2, 2017 | 1066 (16) | 156 (10) | 484 (12) |

although the overall picture does not significantly change with a threshold of $\pm 1$ hours (not shown). The total number of matchups (and individual granules containing matchups) is shown in Table 2.

The AOD is evaluated at the satellite wavelengths used (i.e. bands centred near 470/490, 550, 650/670, and 865 nm, dependent on sensor), as well as 500 nm, as the latter is (along with 550 nm) a commonly-used reference wavelength in aerosol analyses. For the HSRL2 data the available AOD at 355 and 532 nm are interpolated to 470/490 and 500 nm, and extrapolated to 550 nm, using the Ångström exponent (AE, denoted $\alpha$) where

$$\alpha = -\frac{\log \frac{\tau_{\lambda_1}}{\tau_{\lambda_2}}}{\log \frac{\lambda_1}{\lambda_2}} \tag{10}$$

over the wavelength range $\lambda_1 - \lambda_2$ (here 355-532 nm). For 4STAR, up to 12 AOD measurements are available across the relevant wavelength range. Therefore, following Eck et al. (1999) a least-squares fit of all available AODs to a quadratic polynomial is performed and used to calculate the AOD at each wavelength of interest:

$$\log(\tau_\lambda) = a_0 + a_1 \log(\lambda) + a_2 \log(\lambda)^2. \tag{11}$$

Coefficients $a_0$, $a_1$, $a_2$ are calculated on a point-by-point basis. This quadratic formulation is more robust to calibration problems in individual channels, and accounts for the fact that in fine-mode dominant aerosol conditions the relationship between $\log(\tau)$ and $\log(\lambda)$ is not linear but curved, dependent on fine mode particle size (Eck et al., 1999; Schuster et al., 2006). The longer wavelengths are not considered for the HSRL2 comparison to avoid the potentially larger extrapolation errors due to this spectral curvature; likewise, the availability of only two wavelengths means that Equation 11 cannot be applied for HSRL2.

For 4STAR, the uncertainty on an individual matchup is taken as the median of the uncertainties on the spectral AOD used for the fitting in Equation 11 (and is typically around $\pm 0.01$). For HSRL2, the uncertainty is taken as $\pm 0.02$ at 470/490 and 500 nm, and $\pm 0.03$ at 550 nm, to allow for a small extrapolation error. In both cases, the standard deviation of measurements within each satellite footprint is added to this in quadrature, to account for potential spatiotemporal heterogeneity. This latter term is typically 0.01 or smaller, and the total uncertainty is likewise typically much smaller than the estimated uncertainty on the satellite retrievals.



Due to the different flight locations (Figure 5) and potential for different systematic uncertainties in the airborne data between deployments, results are reported separately for 2016 and 2017. The main metrics used to evaluate the AAC retrievals are as follows:

1. The correlation coefficient (R), as a measure of how well the satellite data track the variability of the airborne data. Spearman's rank correlation coefficient is used rather than the more common Pearson's linear correlation coefficient. The reasons for this include the facts that the relationship between airborne and satellite AOD may not be linear, and also that Spearman's correlation is less sensitive to extreme outliers (either sampling-related or retrieval problems) which may be unrepresentative of the behaviour of the data set. While Pearson correlation has historically been the more frequently-used in aerosol data analyses, other fields are increasingly appreciating the use of Spearman correlation for situations where this is better supported by the nature of the data (e.g. the medical literature, such as Schober et al., 2018).

2. The median bias between the data sets, defined satellite-airborne, as a measure of the general offset. Again, medians are more robust to outliers which can skew the means.

3. The median relative bias between the data sets, defined (as above) relative to the airborne data.

4. The root mean square error (RMSE), which is a commonly-reported metric, although is dependent upon the typical level of AOD as well as the presence of outliers.

5. The mean absolute error (MAE), similar to RMSE but less weighted by outliers.

6. The fraction ($f$) of points matching within the total expected level of difference (ED). This ED is taken as the quadrature sum of the expected retrieval uncertainty (Equation 2) and aforementioned airborne uncertainty/variability, under the assumption that these two are independent. If these uncertainties are appropriate, then one standard deviation ($\sim$68 %) of matchups should be in agreement within this bound, and two standard deviations ($\sim$95 %) within twice this bound, etc. Again, however, the spatiotemporal autocorrelation in the observations and limited sample size mean that this is unlikely to be true for this particular set of data. Still, the metric provides a general guideline on how quantitatively similar the estimated uncertainties are to the actual retrieval errors. Note this statistic is not presented for the granule-average comparison, because it is not meaningful for that case.

## 3.3 Validation results

Figure 6 shows one example of instantaneous and granule-averaged results, for the case of MODIS Aqua and 4STAR data in 2016. Here, 15 granules contributed a total of 432 matchups. The bulk of the points in both cases cluster around the 1:1 line. For the instantaneous matchups, there are some outliers, which tend to be retrieved with a large estimated uncertainty; in general, the estimated uncertainty on the satellite retrievals is, as expected, larger than that due to uncertainty and variability in the airborne data. A lot of the scatter is decreased when going to granule-averaged statistics, such that correlation increases and MAE and RMSE decrease. The absolute bias does not change much. Interestingly, the variability on the granule-averaged





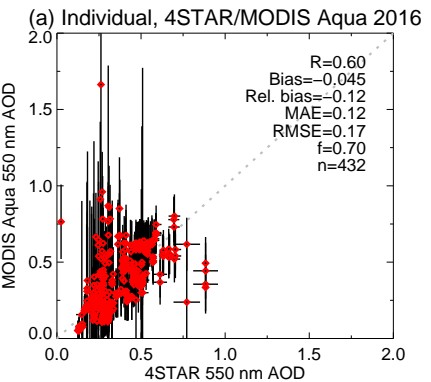
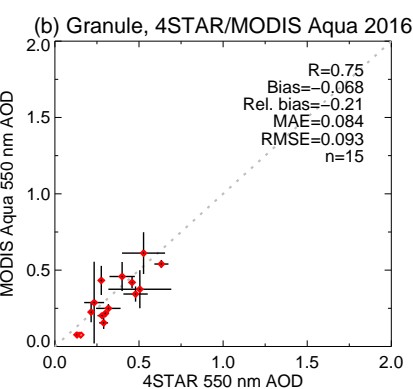

**Figure 6.** Scatter plots and summary statistics for the comparison between 550 nm AOD from MODIS Aqua AAC retrievals and 4STAR data, during ORACLES 2016. Statistics are as defined in the text. Panel (a) shows the comparison for all individual matchups; horizontal and vertical error bars indicate the estimated uncertainties on the 4STAR and satellite retrievals, respectively. Panel (b) shows median 4STAR and MODIS data from matchups obtained within each granule, and horizontal and vertical error bars the standard deviation of matched 4STAR and satellite AOD within each granule, respectively. The 1:1 line is dotted grey.

satellite data (vertical bars in the right panel) tends to be somewhat smaller than the uncertainty on the individual matchups (vertical bars on the left panel). This is likely due to high autocorrelation in the retrieval uncertainties (i.e. an error source on a given retrieval is likely to be very similar to the errors on retrievals adjacent to it), which is a result of the flight-track

sampling of airborne data. This also indicates that, as with many other AOD retrieval algorithms, the bulk of the error is not true random noise but rather locally systematic due to the context (i.e. geometry, atmospheric and surface conditions) of the retrieval. Similar patterns (not shown) are observed for the other satellite sensors and airborne deployments.

In Figure 7, summary statistics equivalent to those presented in Figure 6, but for all wavelengths and satellite/airborne comparisons assessed, are presented. Several statistics (e.g. correlation, $f$) show limited spectral dependence. Others (e.g.

RMSE, MAE, and in some cases the bias) shrink with increasing wavelength, which is expected due to the rapid decrease of AOD of smoke with increasing wavelength. Results for the granule-averaged comparison are often similar to those from the instantaneous comparison (sometimes slightly better, sometimes slightly worse); the same tendencies are seen between satellite sensors and across wavelengths. This also points to the bulk of the errors in the retrieval being contextual rather than truly random (aside from a few individual outlying pixels). The HSRL2 comparison shows smaller difference between

instantaneous and granule-averaged comparison statistics than 4STAR, perhaps due to the generally larger number of matchups, but smaller number of contributing granules, for HSRL2.

Interestingly, the different ORACLES comparison data sets reveal some different patterns. For example, the 2016 data (both 4STAR and HSRL2) indicate near-zero (MODIS) and negative (VIIRS) bias tendencies in the satellite data, while for the 2017 data the biases tend to be more positive. In this sense, the different deployments do not paint identical pictures about the retrieval error characteristics. Recalling the facts that in 2016 4STAR and HSRL2 were on separate aircraft but flying in similar





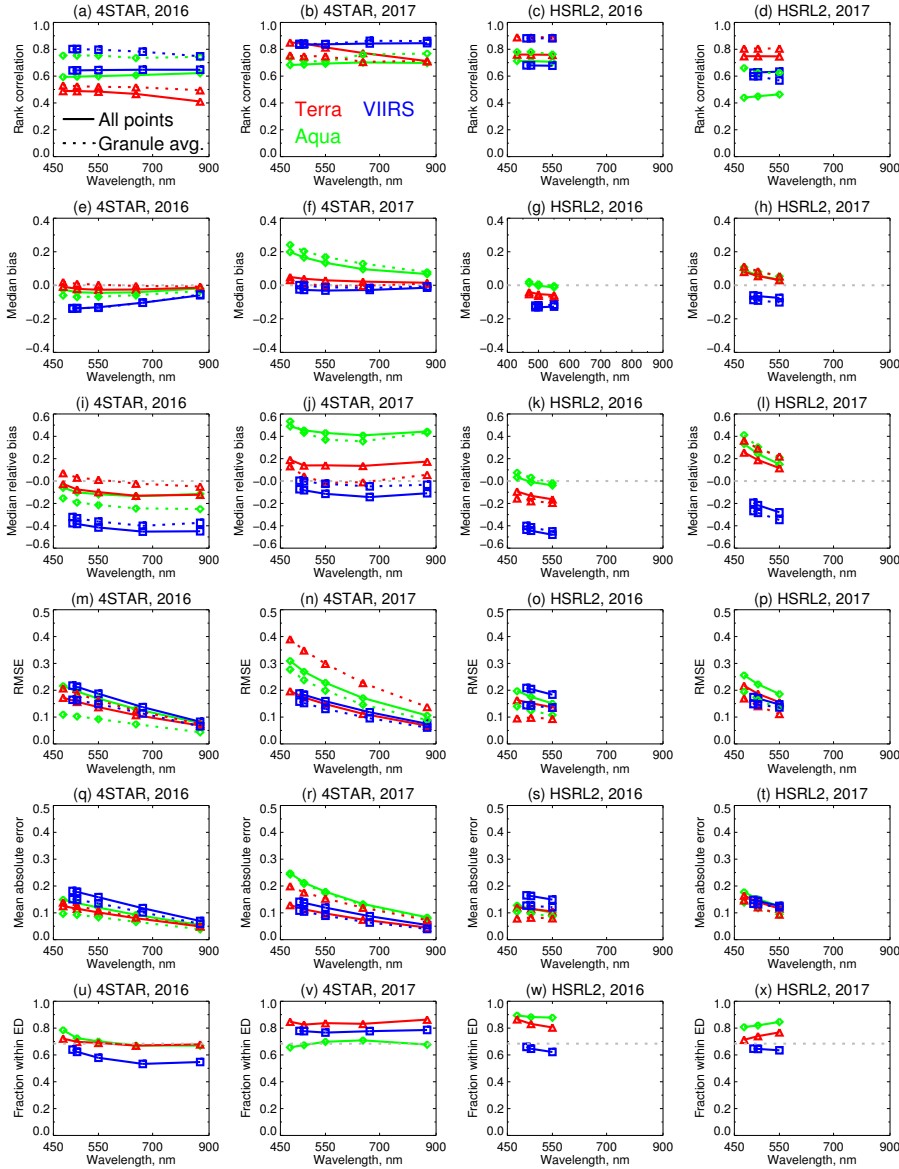

**Figure 7.** Summary line plots of spectral AOD validation statistics. Columns show (left to right) comparisons for 4STAR 2016, 4STAR 2017, HSRL2 2016, and HSRL2 2017. Rows show (top to bottom) rank correlation, median (satellite-airborne) bias, median relative bias, RMSE, MAE, and fraction $f$ agreeing within the ED. In all panels, solid lines denote statistics for all matchups, and dashed for granule-average comparisons. Data for MODIS Terra, MODIS Aqua, and VIIRS are shown in red triangles, green diamonds, and blue squares respectively.

locations and at similar times, while in 2017 they were on the same aircraft and flying over a different region (Figure 5), this suggests that data sets from single deployments may not be providing sufficient sampling of the aerosol/cloud system to fully characterise satellite retrieval uncertainties. The differences might be partially coincidental due to the particular cases sampled





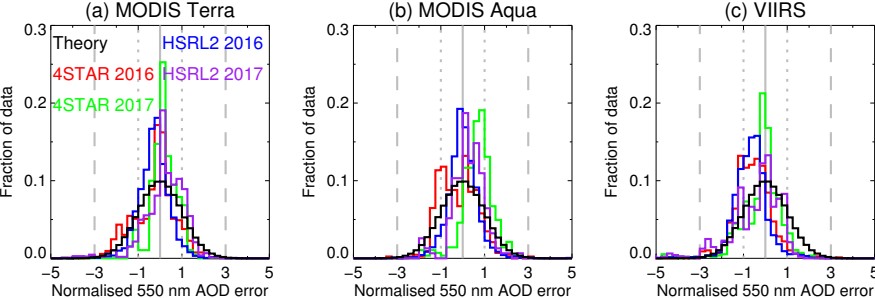

**Figure 8.** Histograms of normalised retrieval error (i.e. actual error divided by expected difference ED) for AOD at 550 nm. Panels show (left-right) data for MODIS Terra, MODIS Aqua, and VIIRS matchups. In all cases matchups from 4STAR 2016, 4STAR 2017, HSRL2 2016, and HSRL2 2017 are shown in red, green, blue, and purple respectively. The black line shows the theoretical Gaussian distribution with mean 0 and variance 1, and dotted/dashed lines indicate $\pm 1$ and $\pm 3$ standard deviations, respectively.

on the flights, or may reflect more persistent differences in the locations of the two deployments; it is difficult to disentangle these two possibilities with the available data. It is therefore cautioned that the validation results presented here may not have sufficient sampling to be comprehensive, and further field campaigns in this region (and others) would be desireable to obtain

a fuller validation of AAC retrievals. Note that the 2018 ORACLES flight tracks followed a similar pattern to those in 2017; the 2018 data are not publicly available at the time of writing and an initial release is expected later in 2019.

The similarity between MAE and RMSE lines in Figure 7 indicates that there are few extreme outliers, as RMSE is sensitive to outliers while MAE is more robust. This is encouraging and provides further evidence that the QA tests (Section 2.3.6) are reasonably successful at removing cases where the forward model is inappropriate. The fraction $f$ of matchups agreeing

within ED is similar to the theoretical value of 68 %, indicating that on average the estimated uncertainties provided by the OE technique (Equation 2) and uncertainty characterisation of the airborne data are reasonable. Note that Equation 2 is applied directly to estimate the uncertainty on AOD at 550 nm, as this is the quantity in the retrieval state vector; this is propagated to the other wavelengths by representing as a fractional uncertainty and scaling by the spectral dependence of AOD assumed in the AAC retrieval. The spectral stability of $f$ (as well as the other statistics) are further evidence that the uncertainty characterisation

and retrieval assumptions are reasonable, although both of these aspects will be assessed in further detail below.

As noted, theoretically the ED should indicate the one standard deviation ($1\sigma$, $\sim$68th percentile) expectation of disagreement between satellite and airborne data. Thus, collectively, the distribution of normalised retrieval error (i.e. the satellite-airborne AOD difference, divided by the ED for each matchup) should approximate a Gaussian distribution with mean 0 and variance 1. A normalised error of $+1$ means that the retrieved AOD was $1 \times$ED higher than the airborne AOD for a particular matchup,

for example. This distribution is assessed for the 550 nm data in Figure 8. The distributions appear reasonable, although tend to peak too strongly near a normalised error of 0, and (particularly for VIIRS) have more negative outliers than expected. Differences between the statistics for the different ORACLES deployments are again also visible. Figure 9 examines this another way, comparing the actual and expected retrieval errors, as a function of ED (in 10 equally-populated bins, in each





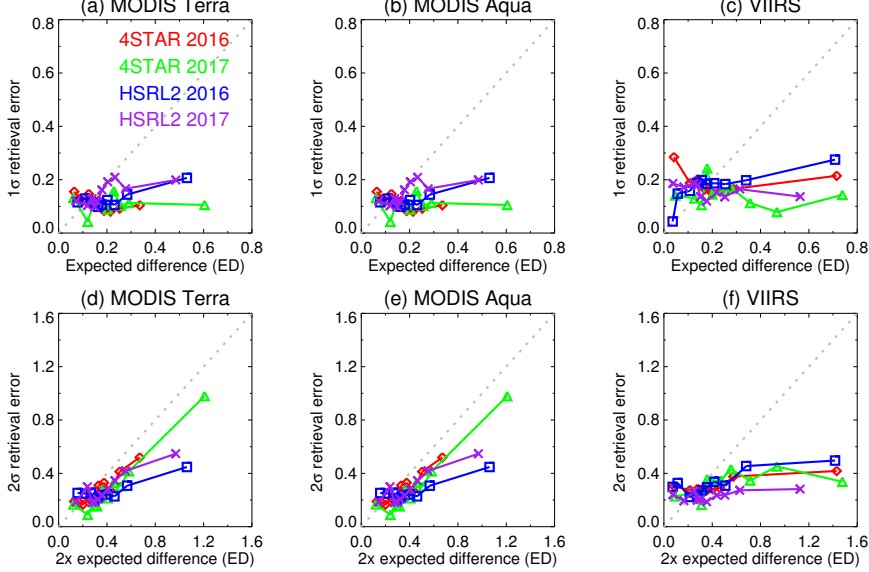

**Figure 9.** Comparison between magnitudes of expected difference (ED) and actual absolute retrieval errors. The top row shows ED (i.e. $1\sigma$ uncertainty) against 68th percentile (i.e. $1\sigma$) retrieval error, binned as a function of ED. The bottom row shows $2\times$ED (i.e. $2\sigma$ uncertainty) against 95th percentile (i.e. $2\sigma$) retrieval error, for the same bins. Panels show (left-right) data for MODIS Terra, MODIS Aqua, and VIIRS matchups. Colours are as in Figure 9. The 1:1 line is dotted grey.

case). Here, the top row compares actual vs. expected $1\sigma$ errors (i.e. 68th percentile of absolute retrieval error in each bin), and the bottom row the same for $2\sigma$ (i.e. 95th percentile) errors. For a perfectly-characterised retrieval system, these points should lie on the 1:1 line. They share a common tendency for underestimating the retrieval error when the ED is low, and overestimating when it is high, with the crossover point being an ED around 0.15-0.2. This latter point (i.e. if a large ED is estimated, it tends to be too large) was also found in the retrieval simulations performed in Sayer et al. (2016). This may be due to nonlinearity in the retrieval system in these conditions, in which case the validity of the OE uncertainty estimate is expected to break down.

The opposite case (i.e. if a very small ED is estimated, it tends to be too small) most commonly occurs when the satellite-retrieved AAC AOD is near zero but the airborne data report an AOD around 0.1-0.15. This suggests that the error budget is missing some component which can be important in fairly low-AOD conditions, perhaps related to calibration uncertainty, the cloud model, or some correlation between forward model error at different wavelengths. From Figure 8, these large negative outliers tend to occur more frequently in VIIRS than in MODIS. This is further supported by quantile-quantile (QQ) plots of the matched data, shown in Figure 10. The QQ plots reveal that for MODIS Terra/Aqua the distributions of satellite and airborne AOD are fairly similar (although satellite AOD are often slightly higher). In contrast for VIIRS it is common for the retrieval to report near-zero AOD a disproportionately high fraction of the time. The reasons for this are not yet known; it is plausible that they are related to limitations of the current cloud mask used (Section 2.3.1). VIIRS also has a broader swath than

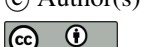



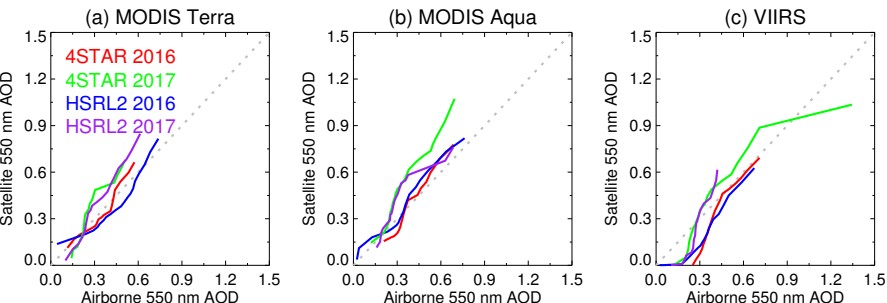

**Figure 10.** Quantile-quantile (QQ) plots comparing distributions of AODs from colocated satellite and airborne measurements, from 5th to 95th percentiles of the matched data. Panels show (left-right) data for MODIS Terra, MODIS Aqua, and VIIRS matchups. Colours are as in Figure 9. The 1:1 line is dotted grey.

MODIS, although retrieval errors as a function of viewing and scattering angles were examined for all sensors and no patterns could be found with the available sampling (not shown). Since the sensitivity to AOD comes largely from the magnitude of spectral darkening across the visible wavelength range, it is also possible that a small calibration of forward model bias is

responsible. Overall, these results indicate that the MODIS-derived AAC record is presently likely to be more reliable than the VIIRS-derived AAC record. Note that in Figure 10 the lines belonging to data for the same year are more similar to each other than the lines for the same instruments (i.e. 4STAR or HSRL2) for different years, further implying that apparent differences in performance are likely related to the specific scenes observed each year.

Nevertheless, the bottom row of Figure 9 shows that the tails of the uncertainty distribution ($2\sigma$ errors) tend to be quantita-

tively better estimated than the ($1\sigma$ errors). This indicates that the current uncertainty estimates do have some quantitative value for identifying retrievals with larger errors. The combination of occassional large positive and negative outliers, and the fact that the ED is somewhat linked to the retrieved AOD (low-AOD cases tend to have a low ED, high-AOD a higher ED) suggests that for calculating daily level 3 aggregate data, medians may be a better option than either simple means or error-weighted means. This is because the AOD fields tend to be fairly spatially coherent, while either a simple or weighted mean may bias

the aggregate.

### 3.4  Evaluation of retrieval assumptions

### 3.4.1  Spectral dependence of AOD

The AOD-dependence of the size distribution in the aerosol optical model assumed in the retrieval (Section 2.2) results in the wavelength-dependence of AOD being a function of aerosol loading. This dependence, illustrated as the AE over the

wavelength range 470-870 nm (cf. Equation 10), is shown in Figure 11. The decline from values near 2 in low-AOD conditions to ~1.7 in high-AOD conditions is a result of the AOD-dependence of Equations 5 and 6.





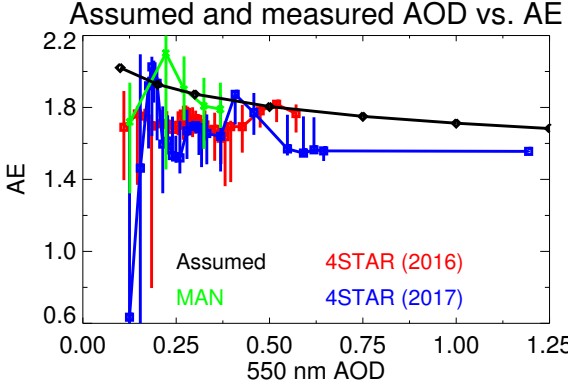

**Figure 11.** AE (470-870 nm) assumed in the retrieval as a function of the AOD at 550 nm (black), together with airborne 4STAR data from the 2016 (red) and 2017 (blue) ORACLES deployments, and the estimated smoke component (see text) of MAN cruises in the region (green). For the 4STAR and MAN data, points and lines indicate bin medians and central 68 % of data, respectively.

These data are compared with two other sources; first is the AE calculated over the same wavelength range (from all available spectral AODs) from the 2016 and 2017 4STAR deployments. These data are then divided into 25 evenly-populated bins as a function of the AOD at 550 nm. The second is four Maritime Aerosol Network (MAN, Smirnov et al., 2009) cruises (Saint Helena 2016, and Research Vessel Meteor 2013, 2015, and 2016) within the region in recent years. These cruises included measurements near the coast, in the open ocean, and under varying levels of smoke influence. The MAN AOD data consist of measurements with a hand-held Microtops Sun photometer, with a typical level of uncertainty around ±0.02 (Knobelspiesse et al., 2004). As they measure total column AOD in cloud-free conditions, they include the contribution from maritime aerosol as well as smoke layers. Here, the maritime contribution is taken as the 20th percentile of all AODs measured in these cruises (0.168, 0.148, 0.112, and 0.093 at 440, 500, 675, and 870 nm respectively), which is then subtracted from the total MAN AOD. The remaining AOD is assumed to be smoke, from which the 550 nm AOD and AE are calculated and the same binning exercise carried out (using only 5 bins due to the small data volume). Using the 20th percentile as a threshold is somewhat arbitrary, although the resulting midvisible AOD is similar to other estimates, and reasonable changes to this threshold result in changes of order ±0.1 to bin-median AE. Failing to remove a maritime component would give a misleading AOD-dependence, as low-AOD conditions would be dominated by the background (low-AE) sea spray aerosol while high-AOD conditions would be more dominated by smoke (but still influenced by the maritime contribution).

All these data exclude points with AOD at 550 nm below 0.1, as AE calculation is highly uncertain when the AOD is low (e.g. Wagner and Silva, 2008), leaving 34397, 14988, and 220 4STAR 2016, 2017, and MAN data points respectively. The variability within these binned data arises from both noise in the AE calculation and real spatiotemporal variability. Note that some of the low-AOD 4STAR data from 2017 are thought to have sampled a small amount of dust, which would explain the anomalously low AE in the lowest bins. HSRL2 data are not used here because of the aforementioned dependence of AE estimates on the wavelength range used; the different wavelength range of HSRL2 (355-532 nm) would be expected to result



in AE values lower by ∼0.5 compared to the 440-870 nm spectral range (Eck et al., 1999; Schuster et al., 2006) and so would be less directly comparable.

Figure 11 shows that the optical model assumed in the retrieval lies on the upper end of the 4STAR and MAN observations. There appears to be a closer match with MAN values, although these are more uncertain than 4STAR due to the subtraction of the estimated maritime contribution. The offset from bin-median 4STAR AE values over much of the AOD range is ∼0.2. The practical implications of an AE overestimate of 0.2 translate to an approximate 3 % overestimate and 10 % underestimate of AOD at the most extreme wavelengths of 470 and 870 nm, respectively, which is somewhat smaller than the total estimate of retrieval uncertainty in most cases. Therefore the spectral dependence of AOD in the aerosol optical model assumed in the retrieval seems reasonable. Using 4STAR data from the 2016 deployment, LeBlanc et al. (2019) observed a general tendency for increasing AE with altitude for AAC cases (ranging from ∼1.5 at 0.5 km to ∼2.0 near 4 km). The data collected were most dense from around 1-2 km, over which the AE was fairly flat around 1.7; altitudes below 1 km or above 3 km, were comparatively poorly-sampled and so possibly less representative. This vertical structure is therefore also a secondary contribution to the overall retrieval uncertainty. The uncertainty and variation within the 4STAR and MAN data are insufficient to determine whether the small decrease in AE with increasing AOD in the assumed optical model is reproduced by these direct-Sun measurements. The 2017 4STAR data do show this decline, although as this draws from a small number of flights it may not be representative.

### 3.4.2 Aerosol SSA

Similarly to the AE case above, the SSA assumed in the AAC retrieval also varies with AOD due to the changing particle size (Equations 5 and 6). This AOD-dependence is fairly small: a range of ∼0.02 at 470 nm and ∼0.04 at 870 nm. This is shown in Figure 12, together with SSA estimated from a range of other sources within this region. These are split by month of year, as previous work (e.g. Eck et al., 2013; Zuidema et al., 2018) has reported a gradual increase in SSA through the burning season. SSA variations are linked to differences in fuel types, ageing, as well as a transition between flaming and smoldering combustion through the burning season resulting in part from meteorological patterns (Zheng et al., 2018).

The first are AERONET inversions from nine sites. Four of these sites tend to sample near-source burning (and cf. Figure 2): Lubango (14.9 °S, 13.4 °E) is in Angola, slightly to the west of fire hotspots and near the coast. Mongu (15.3 °S, 25.2 °E) is in Zambia, surrounded by intense agricultural burning. Mongu ceased operations in 2009 and was replaced by Mongu Inn, in essentially the same location, in 2014. Finally, the Station d'Etudes des Gorilles et des Chimpanzis (SEGC) Lobe Gabon (0.2 °S, 11.6 °E) site is in a remote wildlife research facility in Gabon, northwest of fire sources. SEGC is closer to the latitude of the ORACLES 2017 flight tracks, while the other sites are at latitudes more similar to the 2016 flights. The other five sites tend to sample transported smoke. Ascension Island (8.0 °S, 14.1 °W) is a small island site in the Atlantic, and was the setting for the 2016 LASIC field campaign. Gobabeb (23.6 °S, 15.0 °E) and Henties Bay (22.1 °S, 14.3 °E) are in coastal Namibia, towards the southern end one the main smoke transport pathways. They occasionally also sample local dust sources. Windpoort (19.4 °S, 15.5 °E) is around 400 km to the north of these, closer to the middle of this pathway. Finally, Skukuza (25.0 °S, 31.6 °E) is in eastern South Africa, representing smoke from often similar sources but transported in the opposite





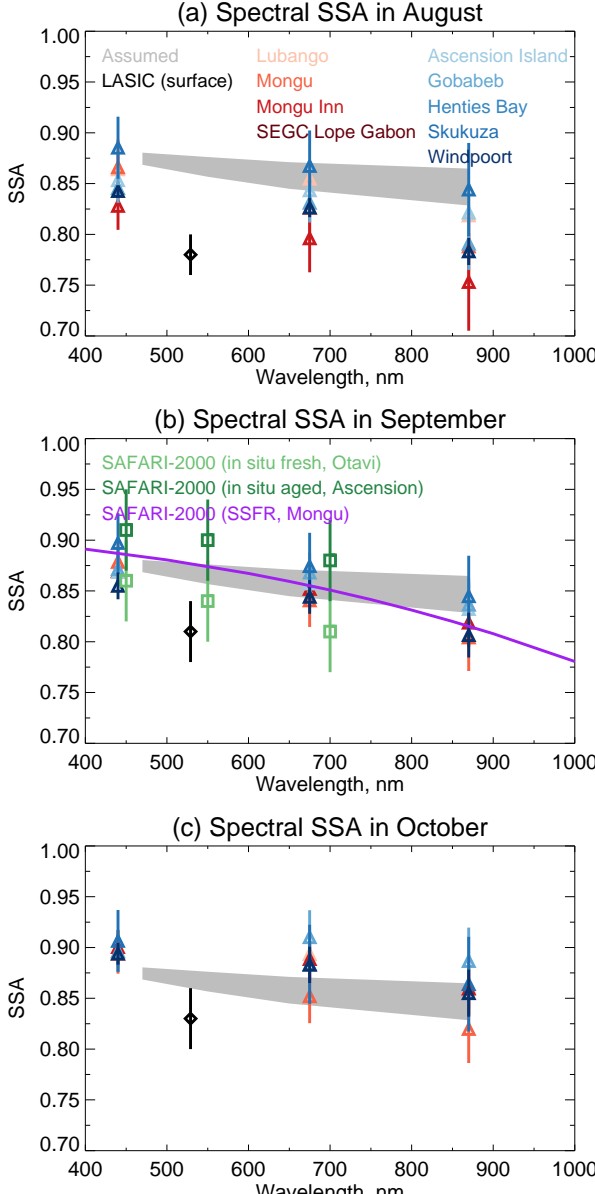

**Figure 12.** Relevant spectral SSA data collected in (a) August, (b) September, and (c) October. The grey shaded region indicates the range of the assumed SSA for midvisible AOD between 0.1 and 1. Mean and standard deviation of AERONET inversions at sites which tend to sample near-source and transported smoke are shown in red and blue triangles respectively. Monthly mean and standard deviation of surface measurements during the LASIC field campaign, reported by Zuidema et al. (2018), are in black diamonds. Airborne in situ measurements from SAFARI-2000 reported by Haywood et al. (2003) are shown for fresh and aged smoke in light and dark green boxes, respectively; airborne remotely-sensed measurements from SAFARI-2000 reported by Bergstrom et al. (2003) are shown in purple.





direction; the aerosol loading here can contain additional sulfate contributions from power plants (Piketh et al., 1999; Eck et al., 2013). Note that these classifications are somewhat fuzzy as given sites can often sample smoke from a variety of ages, due to meteorological patterns which can lead to recirculation of air masses over the continent (e.g. Swap et al., 1996; Tyson, 1997).

Still, the classification and set of sites provide an indication of the regional variation of SSA encountered.

The AERONET data shown are monthly means and standard deviations of version 3 level 2.0 inversions; these level 2.0 data are pre-filtered to remove poor-quality retrievals, as well as retrievals with an AOD at 440 nm<0.4, for which the SSA provided is quantitatively less reliable. Data were further filtered to remove points with AE<1.2, to restrict to smoke-dominated cases, although this had a negligible effect on the results. The uncertainty on the level 2.0 SSA is expected to be ±0.03 (Dubovik

et al., 2000). The bulk of this uncertainty is due to calibration uncertainty, and is therefore expected to be systematic within a given (roughly year-long) deployment (Dubovik et al., 2000; Eck et al., 2013); most of these sites are multi-year records, such that these uncertainties may partially cancel out. Comparisons with the previous version 2 AERONET data (not shown) reveal quantiatively similar climatological results for this region.

Also shown are monthly mean and standard deviation of surface-based estimates made at Ascension Island from the LASIC

field campaign in 2016, reported by Zuidema et al. (2018). These estimated the SSA at 529 nm from nephelometer measurements of aerosol scattering and particle soot absorption photometer (PSAP) measurements of absorption. Zuidema et al. (2018) cautioned that these near-surface measurements (sampling air masses from the boundary layer) may not always be representative of the total column, and noted that Leahy et al. (2007) found that airborne in situ observations from SAFARI-2000 tended to report lower SSA than total-column estimates.

Next, SSA at 450, 550, and 700 nm for two SAFARI-2000 flights from Table 2 of Haywood et al. (2003) are shown. These correspond to flights sampling fresh smoke (the "Otavi plume", flight a790, 13 September) and aged smoke at Ascension Island (flight a794, 19 September). These were computed from size distributions measured by a Passive Cavity Aerosol Spectrometer Probe (PCASP) with an assumed refractive index, resulting in an estimated uncertainty of ±0.04. The Otavi case (but not the Ascension Island flight) also included nephelometer/PSAP measurements which gave a very similar SSA for this case;

Haywood et al. (2003) noted that refractive index assumptions used in the PCASP calculation were informed by the PSAP data. However, more recent work by Ogren (2010) revealed an error in the corrections applied in PSAP data processing (which estimates absorption by measuring transmission through a filter, requiring instrument-specific corrections for flow rate and filter area) which result in a typical underestimate of reported absorption by ∼13 %. For the Haywood et al. (2003) cases, this translates to an overestimate of SSA by ∼0.02, which is within the notional uncertainty but is systematic. It is plausible that

this influences the PCASP results presented here, as they were informed by PSAP data. Note that the newer PSAP analyses such as Zuidema et al. (2018) shown here apply corrections accounting for this error.

SAFARI-2000 results for flight a786 near Mongu reported by Bergstrom et al. (2003), calculated from Solar Spectral Flux Radiometer (SSFR) data, are also shown. The SSFR technique uses measurements of flux above and below the smoke layer, constrained by meteorological data and AATS AOD. The SSA uncertainty on individual wavelengths was ∼0.02 for this case; the best-fit curve is plotted in the Figure. Haywood et al. (2003) also presented results for that flight, and the Bergstrom et al.



(2003) data lie in between the PCASP and nephelometer/PSAP results for that case (not shown); it is not clear whether data from the same parts of this flight were used for those cases, and the age of the smoke sampled is uncertain.

The patterns revealed by these data are in general consistent with previous studies (e.g. Reid et al., 2005; Eck et al., 2013, and references therein). There is a general increase in SSA from August to October (by ∼0.05), and data sampled near source regions tends to be more strongly absorbing than data from aged air masses. The AERONET site at Windpoort is the most strongly absorbing of those placed into the predominantly-transported sites, perhaps because it is closer to the source regions than the others. AERONET monthly mean results from Mongu and Mongu Inn are often similar but can be offset by up to 0.04. The range of interannual variability in SSA at Mongu (not shown) is typically ∼0.05, although some of that would be expected to average out over the time period available (10 years for Mongu, 4 for Mongu Inn). It is therefore possible that the two sites have slightly different error characteristics. The LASIC data are somewhat lower than the others, possibly due to the aforementioned surface sampling.

Within a given month, the AERONET and SAFARI-2000 results tend to span a range of ∼0.06 in the blue spectral region and a larger range of up to ∼0.1 at red and nIR wavelengths, with variabilty tending to decrease through the burning season. The retrieval's SSA assumption (which is centred near 0.875, 0.87, 0.86, and 0.85 at 470, 550, 650, and 870 nm respectively) is in the middle of this range, although the large variability in the reference data suggests large spatiotemporal variability in aerosol optical properties. This implies the potential for spatial/temporal structure in the AAC retrieval error, as the result of the simple assumptions made in the present algorithm version, which may explain some of the differences in bias between 2016 (mostly September) and 2017 (mostly August) flights in Section 3.3. However, Sayer et al. (2016) found that the AAC algorithm was less sensitive to errors in forward model assumptions (such as SSA) in cases of strongly-absorbing aerosols (such as here) than weakly-absorbing aerosols. This is because the sensitivity (darkening) of TOA reflectance to changes in AOD is stronger for more strongly-absorbing aerosols. The Bergstrom et al. (2003) data suggest curvature in the spectral slope of SSA, while the other data sets have insufficient spectral sampling and too large uncertainty to assess whether or not the spectral slope of SSA is curved; the assumed variation is fairly close to linear. The curvature leads to a more pronounced decrease in SSA in the nIR, although since smoke AOD drops off rapidly with increasing wavelength (for an AE of 1.7 as typical in Figure 11, the AOD at 870 nm is about 45 % that at 550 nm) the retrieval is also less sensitive to errors in SSA at this wavelength.

SSA is an inherently difficult quantity to measure. A summary of these results is that SSA is variable in space and time in this region, but the retrieval assumptions are broadly in-family. Analysis of in situ and remotely-sensed SSA data from ORACLES and CLARIFY is ongoing (K. Szpek, personal communication, 2018), but when complete, will be combined with these results to inform possible updates to the optical model, such as considering a seasonally- and/or longitudinally-dependent SSA. This must, however, be balanced against the danger of over-tuning results to a limited data set, which is why the original AERONET-based optical model from Sayer et al. (2014a) used in Sayer et al. (2016) is retained for the present work.



### 3.4.3 Vertical structure

As noted in Section 2.2, the algorithm assumes a cloud $0.3\,\mathrm{km}$ thick with a top altitude of $1.5\,\mathrm{km}$ above surface level, with an overlying aerosol layer $0.5\,\mathrm{km}$ thick with a top height $1\,\mathrm{km}$ above the cloud top. Both cloud and aerosol are assumed to be

vertically homogeneous.

Initial analyses of aerosol/cloud altitudes in this region were generally performed by CALIOP data; Rajapakshe et al. (2017) provide a summary of some of these results. However, a combination of sensor design and algorithmic limitations mean that CALIOP has been shown to overestimate the bottom of optically-thick aerosol layers, leading both to an underestimate of the above-cloud AOD and also an overestimate of the gap between cloud top and the overlying aerosol layer bottom (e.g. Liu et al.,

2015; Rajapakshe et al., 2017). This is due to attenuation of the laser signal through the layer meaning that the returns from the bottom portion can be too weak for the layer detection algorithm to work correctly. Improvements in the recent CALIOP version 4 data relating to calibration, lidar ratios, and layer detection (Liu et al., 2018) mean that this issue has been slightly ameliorated, although not fully bypassed.

Recently, Rajapakshe et al. (2017) performed a similar analysis of aerosol/cloud vertical structure in this region, using both

CALIPSO and CATS data during the 2015-2016 burning seasons (July-October). As the CATS lidar used a wavelength of $1064\,\mathrm{nm}$, the aerosol signal is generally somewhat weaker than at the $532\,\mathrm{nm}$ used by CALIOP, which further lessens the impact of the attenuation issue on layer detection. Overall, they found CATS reported liquid cloud top heights around 1-$1.5\,\mathrm{km}$, typical separations between cloud top and aerosol layer base height $0.25\text{-}0.5\,\mathrm{km}$, and aerosol layer top heights around $3.5\text{-}4.5\,\mathrm{km}$ (for a geometric thickness of $1.5\text{-}3.5\,\mathrm{km}$). The ranges quoted here arise from longitudinal gradients: as the layer

moved West from the coast of Africa into the Atlantic, they found decreases in aerosol top height and increases in cloud top height. Thus, nearer the coast the separation between aerosol and cloud was larger. Meridionally, Rajapakshe et al. (2017) found higher cloud tops and aerosol layer bases nearer the Equator than toward the southern end of the study region, although the separation between the two layers was relatively constant. One limitation of that study was that to decrease solar noise only nighttime CATS data were used, but as these are large-scale features, and CALIPSO day/night differences were not large (aside

from known detection sensitivity issues), it is plausible that these nighttime results also hold for daytime measurements. Note that Rajapakshe et al. (2017) did not examine cloud geometric thickness, although the AAC retrieval algorithm presented here is likewise fairly insensitive to that for opaque clouds.

In light of this, the assumptions made in the AAC retrieval algorithm presented here seem reasonable, although refinements might consider a longitudinal variation of vertical structure and expanding the geometric thickness of the aerosol layer. Never-

theless, Sayer et al. (2016) found that the algorithm was less sensitive to this assumption than other error sources such as SSA assumptions. Jethva et al. (2018) do use a CALIPSO-based climatology of aerosol height data in their OMI data set (but not cloud height or aerosol geometric thickness), which is helpful as retrievals using OMI's UV wavelengths are more sensitive to vertical structure assumptions than the visible/nIR bands used here.



## 4  A 20-year record from SeaWiFS, MODIS, and VIIRS

### 4.1  Time series

This Section briefly examines spatiotemporal patterns in the 20-year record obtained by applying the AAC retrieval algorithm

presented here to the four satellite sensors. A broader study comparing results against other satellite AAC AOD/COD data products is planned for the future. First, monthly time series of the retrievals, and other relevant satellite data sets (using the most recent versions available), are shown in Figure 13. These are constructed by averaging daily data over the green box (25 °S-0 °N, 15 °W-15 °E) in Figure 2, which corresponds to the core of the stratocumulus cloud deck and the main flight region for ORACLES deployments, and then computing monthly averages from these.

Panel (a) shows time series of UVAI from two data records: the multisensor (MS) UVAI data set version 1.7 combines TOMS, GOME, SCIAMACHY, and OMI observations, dividing each of the fairly coarse-resolution sensor pixels into several subpixels to produce a long-term (starting 1978) data set with consistent spatial resolution (Tilstra et al., 2012, 2013). Also shown are the latest version 1.8.9.1 OMI UVAI data, described by Torres et al. (2018). Although this is a shorter record, this latest OMI data version updates the UVAI calculation to decrease variations associated with changes in solar/sensor geometry and particle

shape which influence apparent seasonality. Panel (b) provides the fraction of days within each month where the box-average UVAI was above 0.75, a subjective but reasonable threshold (e.g. Tilstra et al., 2013; Torres et al., 2018) for the presence of non-negligible levels of absorbing aerosols. Panels (c) and (d) present the AAC AOD from the algorithm presented here, and the total column (i.e. from cloud-free scenes) AOD from the MODIS Dark Target (DT) Collection 6.1 over-water algorithm (Levy et al., 2013). Panel (e) provides an estimate of the below-cloud AOD from MODIS, by subtracting the AAC data in panel

(c) from the total AOD in panel (d). Strong caution is required in this as the DT and AAC algorithms are independent and have different error characteristics, although it provides a crude proxy for the relative partitioning of aerosol above and below cloud level. Finally, panel (f) shows fire counts from the cloud-corrected overpass-corrected MODIS Collection 5 (Giglio et al., 2003, 2006) data set (MOD14CM1/MYD14CM1). Unlike the other quantities, the fire counts represent total detections rather than an average, and the longitude range is shifted to cover the source region 10-40 °E (as no fires occur over ocean). The fire data

are also not presently available for the full MODIS record lengths.

In all of these time series (aside from the below-cloud AOD estimates in panel e), the annual cycle of fires and associated emissions, strongest from June-September, is evident. Interannual variability is comparatively limited, but generally consistent between data sets. A smaller secondary peak from December-February is also seen, which is likely linked to a combination of Sahelian fires and dust transport (Pandithurai et al., 2001; Ben-Ami et al., 2009). MODIS Terra fire counts are around a factor

of 5 lower than those observed by Aqua; this pattern was observed in multiple global source regions by Giglio et al. (2006), and ascribed to diurnal variations in fire activity. Table 3 shows the correlation between each of the time series in Figure 13 and the four AAC AOD data sets. Correlation coefficients are high (0.78-0.94), and show small variability between the four AAC AOD records generated here. Due to the small number of points in the time series, the differences in correlation coefficients between sensors are not statistically significant. Likewise, it is not possible to state robustly which quantity is most strongly correlated with the retrieved AAC AOD. These results indicate that these quantities may provide a useful proxy for the amount





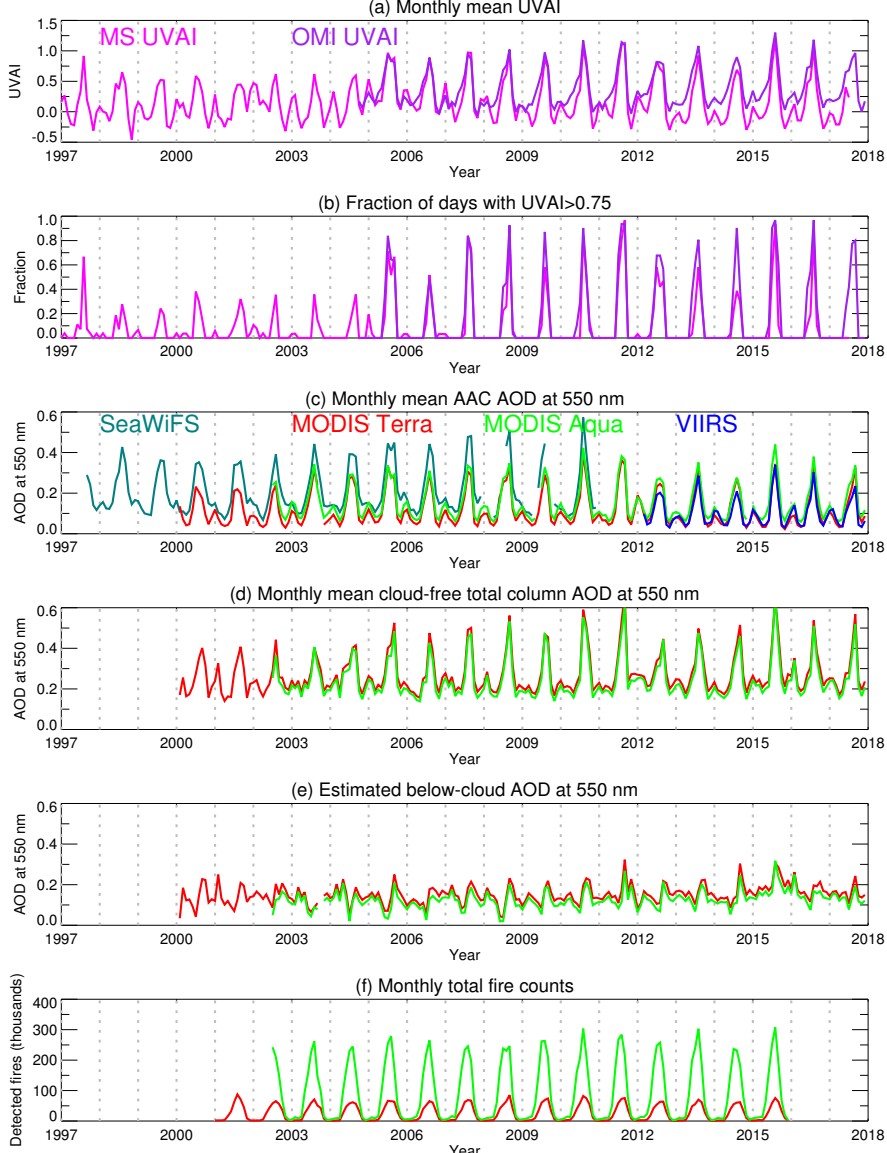

**Figure 13.** Monthly time series of various satellite data sets over the green box (25 °S-0 °N, 15 °W-15 °E) in Figure 2. Panel (a) shows the monthly mean UVAI from the multi-sensor (MS) and OMI data sets, and (b) the fraction of days in each data set where the box-averaged UVAI is over 0.75. Panel (c) shows the mean AAC 550 nm AOD using the algorithm presented in this work, applied to SeaWiFS, MODIS Terra/Aqua, and VIIRS measurements. Panel (d) is a time series of monthly mean total column (cloud-free) over-water 550 nm AOD from the MODIS Terra/Aqua DT data sets. Panel (e) is the difference between total column and above-cloud AOD (i.e. d-c), estimated for MODIS Terra and Aqua. Panel (f) shows monthly total corrected fire counts from MODIS Terra and Aqua (box shifted 25 °E from the others). Throughout, MS data are shown in magenta, OMI in purple, SeaWiFS in teal, MODIS Terra in red, MODIS Aqua in green, and VIIRS in blue. Months with fewer than three contributing days are excluded.





**Table 3.** Correlation coefficients between monthly mean AAC 550 nm AOD and other time series shown in Figure 13.

| Pairing | SeaWiFS | MODIS Terra | MODIS Aqua | VIIRS |
|---|---|---|---|---|
| MS UVAI | 0.84 | 0.91 | 0.93 | 0.91 |
| OMI UVAI | 0.92 | 0.92 | 0.93 | 0.89 |
| Fraction MS UVAI >0.75 | 0.84 | 0.86 | 0.88 | 0.89 |
| Fraction OMI UVAI >0.75 | 0.93 | 0.92 | 0.94 | 0.92 |
| MODIS Terra total column AOD | 0.90 | 0.88 | 0.91 | 0.89 |
| MODIS Aqua total column AOD | 0.91 | 0.90 | 0.91 | 0.91 |
| MODIS Terra below-cloud AOD | 0.13 | 0.069 | 0.14 | 0.23 |
| MODIS Aqua below-cloud AOD | 0.062 | 0.034 | 0.098 | 0.25 |
| MODIS Terra fire counts | 0.91 | 0.85 | 0.86 | 0.78 |
| MODIS Aqua fire counts | 0.90 | 0.88 | 0.88 | 0.84 |

**Table 4.** Comparative statistics for monthly mean AAC 550 nm AOD between the four AAC data sets generated in this work. Offsets are defined subtracting the second indicated sensor from the first.

| Pairing | Correlation | Median offset | RMS difference |
|---|---|---|---|
| SeaWiFS/MODIS Terra | 0.95 | 0.077 | 0.097 |
| SeaWiFS/MODIS Aqua | 0.96 | 0.052 | 0.078 |
| MODIS Terra/Aqua | 0.99 | -0.027 | 0.032 |
| MODIS Terra/VIIRS | 0.96 | 0.0051 | 0.029 |
| MODIS Aqua/VIIRS | 0.97 | 0.040 | 0.052 |

of aerosol transported above clouds, if AAC retrievals are not available. However the strength of the relationships might not hold for other regions where aerosol and cloud properties covary differently.

The exception is the estimated below-cloud AOD, which is only very weakly correlated with the above-cloud AOD. This might imply that very little of the smoke is transported within the marine boundary layer, which is generally consistent with the discussion in Section 3.4.3. However, as mentioned previously, due to large uncertainties caution should be used in interpreting these data. The mean and standard deviation of below-cloud 550 nm AOD estimated from MODIS Terra and Aqua are 0.15±0.05 and 0.13±0.04 respectively, which is only slightly larger than ship-based measurements of AOD in maritime environments without significant continental influence (Smirnov et al., 2009, 2011).

## 4.2 Spatial patterns and offsets

Figure 13 also shows offsets between the AAC retrievals, with SeaWiFS the highest and VIIRS the lowest. This is consistent with the validation results in Section 3 (aside from the SeaWiFS mission which ended in 2010 so cannot be directly validated





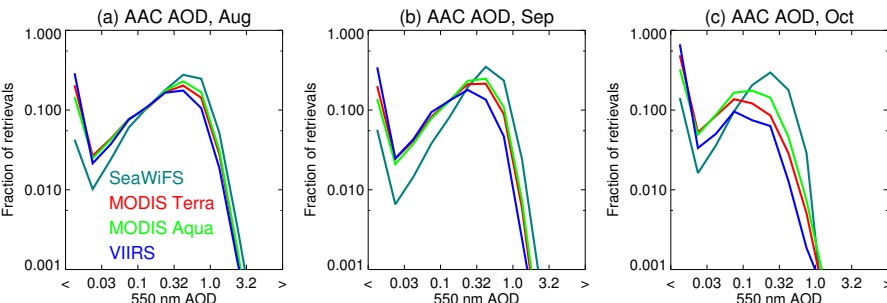

**Figure 14.** Histograms of AAC 550 nm AOD retrieved over the green box (25 °S-0 °N, 15 °W-15 °E) in Figure 2, aggregated from level 2 retrievals over the full satellite records processed, during the ORACLES campaign season. Panels show data for August, September, and October. Throughout, SeaWiFS data are shown in teal, MODIS Terra in red, MODIS Aqua in green, and VIIRS in blue.

with ORACLES data). Table 4 quantifies the consistency between these time series, revealing very high correlation coefficients (0.95-0.99) despite these offsets. Note that Pearson correlation coefficients are calculated for this instance, as the data sets are notionally inferring the same quantity using a similar technique and should be subject to the same causes for outliers (e.g.

extreme events in a given month). Further, Figure 14 shows histograms of the AOD retrieved by all four sensors from August to October. In all cases a significant fraction of the data retrieve near-zero AOD, and have a secondary roughly lognormal distribution of nonzero AOD. All show the decline in AOD (shift to the left) from August to October. But the SeaWiFS and VIIRS histograms are shifted to the right and left respectively, compared to the others. Figure 15 shows that these offsets are in general found across the broader spatial domain, with the four sensors reporting consistent spatial and temporal patterns of

AOD. As well as the main smoke plume in the ORACLES domain, a secondary river of smoke outflow into the southern Indian Ocean, is seen peaking in September. This featured was also observed by Jethva et al. (2018) and Kar et al. (2018) using OMI and CALIOP data, respectively, and is consistent with known transport patterns (Swap et al., 2003). AOD magnitudes are more different over land, although due to lower cloud cover the data volume is significantly lower and so sampling differences may dominate.

While pixel selection and differences in sensor resolution likely also contribute, the shifts in histogram shape may be plausibly ascribed to uncertainties in the absolute calibration of the sensors. As the aerosol signal is small compared to that of the underlying cloud, a spectral bias in calibration or the retrieval forward model could lead to a systematic bias in the retrieved AOD. This is also an issue with clear-sky AOD retrieval algorithms, e.g. despite identical algorithms there are known systmatic offsets in AOD retrieved from several algorithms between MODIS Terra and Aqua (Levy et al., 2013; Sayer et al., 2015b).

Recent work by Chang et al. (2017) used stable ground sites and identified relative offsets of up to around 2 % in the calibration of these two sensors, which is thought to be responsible for these differences.

As noted previously, SeaWiFS was calibrated vicariously against ground-based data as described by Franz et al. (2007). In brief, this method assumes that the calibration at the 865 nm band is correct and then adjusts the gain of the other bands such that water-leaving radiance retrievals at this site are unbiased. While an effective method for the ocean colour applications




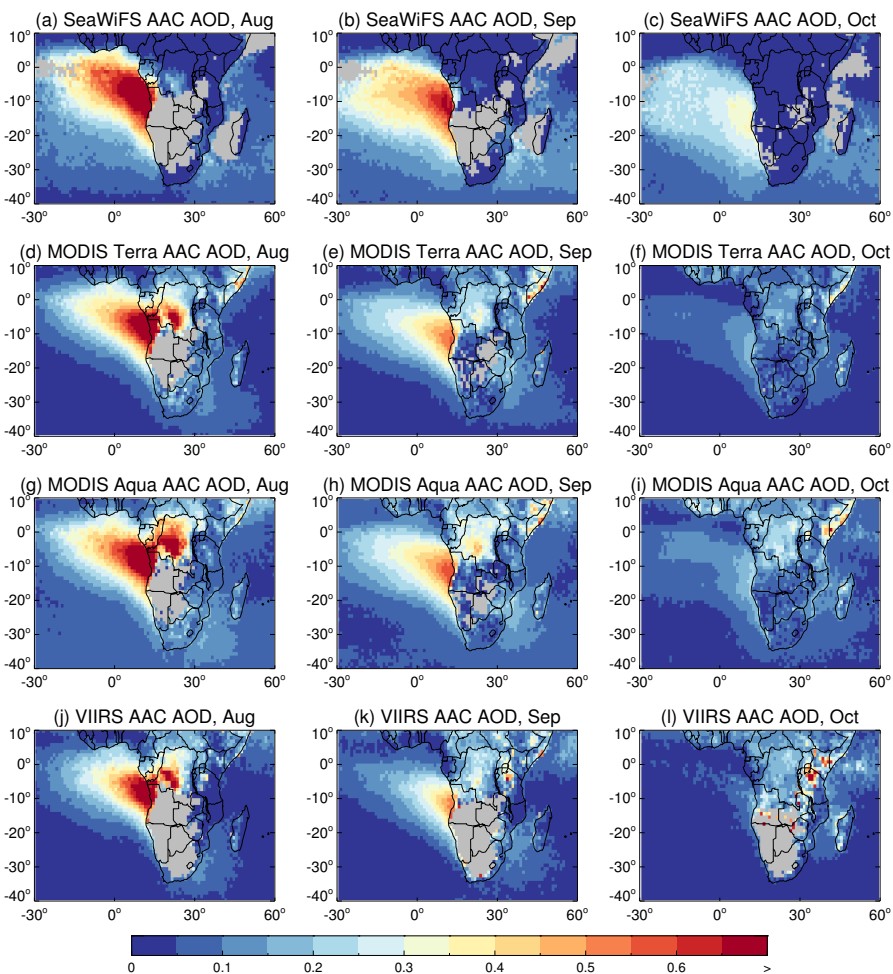

**Figure 15.** Multiannual monthly mean maps of AAC 550 nm AOD. Columns show (left-right) data for August, September, and October. Rows show (top-bottom) data for SeaWiFS, MODIS Terra, MODIS Aqua, and VIIRS. Grid cells with fewer than five years contributing are shown in grey.





which were the main focus of SeaWiFS, this technique has two main disadvantages for others: first, the untested assumption that the 865 nm band is unbiased, and second, that the process propagates errors in the ocean colour retrieval atmospheric correction (relating to e.g. aerosol and trace gas absorption assumptions) into the derived vicarious gain. The latest coefficients

used here apply scaling factors of 0.982, 0.9948, and 0.9648 to the SeaWiFS 490, 550, and 670 nm bands respectively, i.e. tilting reflectance downwards at shorter wavelengths compared to 865 nm, which is the direction which would increase the retrieved AAC AOD. If either of the limitations described above are important, this could explain the rightward shift of the SeaWiFS histograms and positive offset seen in the data. Recent work supports these possibilities. Kahn et al. (2016) found spectral biases in the SeaWiFS atmospheric correction, which is used in the vicarious calibration, and Voss and Flora (2017) illustrated

that simplifications in current water-leaving radiance processing in the reference data used for the vicarious calibration process lead to small spectral biases.

    The VIIRS data used here were cross-calibrated against MODIS Aqua as described in Sayer et al. (2017), consistent with the main VIIRS Deep Blue data processing. This applied corrections of 0.992, 0.956, 0.941, and an average of 0.963 (with some small temporal dependence) to the 490, 550, 670, and 865 nm bands respectively. This scaling would be expected to

decrease the retrieved AAC AOD compared to the uncorrected case. The uncertainties on these corrections were estimated to be ∼0.5-1 %, and similar results were found using analyses of cloudy scenes (K. Meyer, personal communication, 2018). From the validation results here is is plausible that there is a residual spectral bias in the derived calibration which is leading to biased above-cloud AOD, although Sayer et al. (2017) did find that applying this cross-calibration improved clear-sky AOD retrievals, but there was a residual spectral dependence to the AOD bias.

It is difficult to say from the available validation data which data set is closest to the truth. However it seems reasonable to assume that adopting a consistent calibration method for the sensors - whether cross-calibrating against a single satellite reference, or calibrating vicariously against ground targets - may improve the consistency of the time series generated. Trace gas absorption corrections can manifest in a similar way to calibration issues, as they are systematic adjustments to bands. Differences between spectroscopic data bases or correction parametrisations can also lead to offsets in retrievals (Patadia et al.,

2018), so it is also important that these are updated as better spectroscopic measurements or atmospheric reanalyses become available.

## 5   Conclusions

The ORACLES field campaign and others have provided a wealth of valuable information for the evaluation and refinement of AAC retrieval algorithms for smoke in the south-eastern Atlantic Ocean. This study has detailed updates to an AAC retrieval

algorithm intended for eventual incorportation into the Deep Blue satellite aerosol data product suite, and then evaluated this algorithm, chiefly using airborne data collected during the 2016 and 2017 ORACLES deployments. This builds on the initial algorithm presented and evaluated with SAFARI-2000 field campaign data by Sayer et al. (2016), providing the largest-scale validation possible to date, and can further be supplemented by future analysis of ORACLES 2018 and CLARIFY data, as





well as AAC retrievals from airborne polarimeters (e.g. Xu et al., 2018) from all three ORACLES deployments, when these are finalised and become available.

One of the key drives behind the development of this algorithm was to extend coverage of Deep Blue aerosol data products

to include AAC cases and thereby fill in some systematic gaps in these global data sets. The algorithm was developed with this in mind, explaining the choice of spatial resolution, as well as the spectral range of bands used (470-870 nm), as the SeaWiFS, MODIS, and VIIRS sensors which provide the core of the Deep Blue multi-sensor data product suite all include these bands. The validation and time series results reveal a reasonable degree of consistency in the resulting data sets, although with some offsets which are likely due to small systematic calibration differences. Calibration assessment and correction (for

both absolute calibration and on-orbit degradation) remain a challenge to creating consistent multi-sensor data sets (for AOD and other quantities), and small AOD offsets can persist despite similarities in revealed seasonal and interannual variability. This points to the need for continued traceable calibration, with quantified uncertainties, for satellite measurements in the solar spectrum. Ideally this might be achieved on-orbit, as has been done using hyperspectral data for the thermal infrared (e.g. Veglio et al., 2017), in order to enable consistent cross-calibration of multiple instruents against a high-quality reference.

Similar spectral bands are also present on the new generation of geostationary sensors launched in recent years, including the Korean Geostationary Ocean Color Instrument (GOCI), Japanese Advanced Himawari Imager (AHI), and US Advanced Baseline Imager (ABI). GOCI and AHI provide coverage over much of Asia and the Pacific. This includes some important AAC systems, such as springtime biomass burning smoke in south-eastern Asia transported above stratocumulus cloud decks (Hsu et al., 2003; Tsay et al., 2013; Lin et al., 2014). The ABI domain covering much of the Americas and surrounding oceans

samples AACs less frequently, although Jethva et al. (2018) report cases above stratocumulus decks within the ABI domain from 10-30 % of the time in certain months. It would therefore be advantageous to apply Deep Blue and this AAC algorithm to those sensors, improving knowledge of the diurnal cycle of AACs.

The algorithm is complementary to other AAC quantification approaches. No existing sensor is perfect for AAC quantification. For example, passive approaches such as this algorithm and that of Meyer et al. (2015) or Jethva et al. (2018) provide

broad-swath spatial coverage unavailable from active lidar, however they cannot reliably quantify the vertical profiles of the aerosols and clouds (even though, as noted previously, current CALIOP algorithms have a known overstimate of aerosol layer bottom height). UV approaches such as Jethva et al. (2018) can provide a less ambiguous detection of AACs than visible approaches; however, they become quantitatively more sensitive to assumptions made about aerosol optical properties and vertical structure (Hsu et al., 1999), and also (to date) are only available from coarser-resolution sensors, which limits applicability

in areas of smaller or more broken clouds. The algorithm presented here has many similarities to the MODIS-based algorithm of Meyer et al. (2015). That work also provides a retrieval of cloud effective radius (CER), as it is primarily aimed as a regional bias-correction to cloud optical property retrievals. That, however, requires spectral bands which are not present on the SeaWiFS sensor, and the aforementioned goal of this Deep Blue effort is consistency where practical. Sayer et al. (2016) also noted that CER retrieval was not the main contributor to AOD retrieval error for this type of algorithm, so its neglect likely does

not significantly impact the quality of the retrieved aerosol data. The Meyer et al. (2015) approach is also at present applied only over water, while this algorithm (and that of Jethva et al., 2018) can also be applied over land. Multiangle polarimetry,





to date represented (in a spaceborne sense) by POLDER, provides an additional dimension of measurement over intensity-only sensors, which has been applied successfully to AACs (e.g. Waquet et al., 2013; Peers et al., 2015). These have also, however, suffered from coarser spatial resolutions and calibration/polarimetric accuracy limitations. Based on current airborne

prototypes (e.g. Xu et al., 2018), the next generation of spaceborne polarimeters are expected to improve upon POLDER's capabilities in this regard. Interestingly, multiangle intensity-only observations such as from the Multiangle Imaging Spectro-radiometer (MISR) have not been used for the study of AACs, although MISR's spectral range encompasses the bands used in the algorithm presented here, and with multiangle capabilities the sensor should be able to quantify them more robustly than single-view sensors like MODIS or OMI. A further study is planned to compare available AAC data products in more detail

and assess their results in light of sensor strengths and limitations.

Overall, the validation and comparison exercise has revealed that the AAC algorithm presented here performs roughly within expectations. Specific areas for potential refinement have been identified, including sensor calibration and potential spatial/temporal adjustments to assumed aerosol/cloud optical properties and structure. These refinements would be expected to improve the consistency between the different sensors to which the algorithm has been applied, and reduce some sources of

systematic uncertainty at certain times and locations. Moving forward to a global application would require the development of equivalent appropriate assumptions globally, which can be done by leveraging climatologies of vertical structure from lidar (spaceborne or ground-based), and representative aerosol optical properties from AERONET and potentially global model-based climatologies (e.g. Kinne et al., 2013). In the meantime, the AAC data set generated in this work is available for interested researchers.

Unfortunately, the available validation data for these algorithms remains highly sparse. The results here suggest that the available ORACLES flights, while a significant important milestone and far ahead of the characterisation of other AAC systems, may not yet represent sufficient sampling to provide a robust regional validation. The available validation from field campaigns in other regions (including for example eastern Asia, as well as downwind of dust source regions in northern Africa and the Arabian Peninsula) is much more limited. Aircraft observations are a powerful tool to provide data-rich, thorough

characterisation of sampled air masses. Such campaigns could be supplemented by instrumentation carried upon unmanned aerial vehicles (UAVs), a technology which has advanced greatly in recent years (for a recent review see Villa et al., 2016). Frequent launches of instrumented UAVs, combined with geostationary satellites, could provide an important temporal sampling component to further refine understanding of the processes influencing the evolution of these systems. It is critical, both for answering science questions about the role of AACs in the Earth system, and for a robust quantitative validation of spaceborne

AAC data sets, that such observations continue to be made.

*Data availability.* The Deep Blue aerosol-above-cloud retrieval data shown in this work are available upon request to the authors, in advance of their eventual incorporation to the main Deep Blue data products. Links for the input and evaluation data sources used in the analysis are given in the Acknowledgements.



*Author contributions.* AMS developed the algorithm, performed the analysis, and wrote the manuscript. NCH is the PI of the Deep Blue aerosol project; NCH, JL, and WVK are the other key developers for the DB and SOAR clear-sky aerosol retrieval algorithms and provided guidance. SCT provided scientific suggestions and general guidance through the project and algorithm development. SB, MAF, and RAF

were involved in HSRL2 data collection, processing, and analysis. MK, SL, KP, JR, MSR, and YS were involved in 4STAR data collection, processing, and analysis. All authors contributed to reviewing and editing the manuscript.

*Competing interests.* The authors declare no competing interests.

*Acknowledgements.* This research was funded by NASA's Radiation Science Program, managed by Hal Maring. More information about Deep Blue is available from https://deepblue.gsfc.nasa.gov. ORACLES field campaign data and further information are available from https:

//espo.nasa.gov/oracles. ORACLES investigators and their team members are thanked for their considerable efforts into planning, executing, and archiving the results of the ORACLES field campaign deployments. SeaWiFS data are available from https://oceancolor.gsfc.nasa.gov. MODIS and VIIRS atmospheres data are available from https://ladsweb.modaps.eosdis.nasa.gov/. NASA GMAO are thanked for GEOS-5 FP-IT and MERRA2 reanalysis data, available from https://gmao.gsfc.nasa.gov. Crystal Schaaf (UMB) is thanked for useful discussions about the MODIS albedo products; the MCD43GF data used in this study are available from https://www.umb.edu/spectralmass. AERONET

and MAN data are available from https://aeronet.gsfc.nasa.gov; Alexander Smirnov (SSAI) and the AERONET/MAN investigators are thanked for the creation and stewardship of these data records, and useful discussions concerning data from the region of interest. MODIS fire and OMI data are available from https://earthdata.nasa.gov/about/daacs/daac-ges-disc, and MS UVAI data from http://www.icare.univ-lille1. fr. The VIIRS Atmospheres SIPS at the University of Wisconsin are thanked for assistance obtaining and processing VIIRS data. Richard Frey (University of Wisconsin), Steve Platnick (NASA GSFC), and Kerry Meyer (NASA GSFC) are thanked for access to and advice about

the VIIRS cloud mask data product. Tom Eck (USRA), Kate Szpek (UK Met Office), and Connor Flynn (PNNL) are thanked for useful discussions about remote and in situ measurements of aerosol SSA in the study region.



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
