# Peer review of "Two decades observing smoke above clouds in the south-eastern Atlantic Ocean: Deep Blue algorithm updates and validation with ORACLES field campaign data"

_Atmospheric Measurement Techniques, 2019_

## Referee Comment (RC1) · Anonymous Referee #1 · 19 Mar 2019

Main comments

(1) This paper presents a unique 20-year time series of AOD of aerosols-above-clouds in Southern Africa, with an extensive validation. A number of different satellite instruments are used to create a long time series of AAC AOD, which is useful for climate analysis. Especially the validation has been performed extensively and carefully. Figure 7 shows comprehensively the validation of the results. This interesting and important paper deserves publication. The paper is a pleasure to read: it is well written and has a clear structure. The figures are clear and informative. There is a good intro-

duction with good referencing. The methods that are used are sound. The discussion includes the important aspect of calibration, which is different for each instrument used.

(2) In the optimal estimation retrieval scheme which is used, also the error on the AOD is retrieved. It would be informative to give with the AOD map also the AOD retrieval error map, e.g. in Figure 15.

(3) Several satellites are used with different overpass times. Is there an effect of the satellite overpass time, because of diurnal variation of the AAC AOD?

Specific and textual comments

- Caption Figure 2: To which period do the data refer? Please correct: \unit nm.

- p. 6, l. 26: E_lambda should be defined perpendicular to the solar direction, and at 1 AU. In principle, E_lambda should hold at exactly the same time as the radiance measurement.

- Eqs. 7-8 on p. 11, and text below the equations: For these brightness tests, the reflectance becomes a sun-normalized radiance. Does this translation from reflectance to radiance only hold for specific SZA range? Please give the angular range.

- p. 13, l. 10: superfluous bracket after 10 m.

- p. 13, l. 14: do you also correct for NO2, which absorbs in the blue range ?

- p. 16, l. 8: then > than

- Figure 4: at which wavelength does this AOD hold?

- p. 22, l. 2: Please give a reference for these main metrics.

- p. 22, l. 18: please give an equation for f, in which it is related to S in Eq. 2.

- p. 37, l. 11: featured > feature

- Figure 15: Multiannual is too vague. Please give the time period for which these data

hold. You could give this information in a table, together with the overpass times of the various satellites.

- p. 39, l. 30: incorporation

- p. 40, l. 8-9: could these differences be due to different periods used, or different time of day?

- p. 40, l. 13-14: on board calibration using the sun and lamps is used for GOME type sensors, OMI, etc.
* * *

---

## Referee Comment (RC2) · Anonymous Referee #2 · 19 Apr 2019

The goal of the paper by Sayer et al is to provide an update of the aerosol above cloud (AAC) algorithm that retrieves above cloud AOD and liquid COD. This algorithm can then be applied both over land and ocean for sensors such as SeaWiFS, MODIS, and MODIS. The paper also evaluates the results of the algorithm from 2016 and 2017 from the (4STAR/HSRL2) ORACLES filed campaign.

The paper is generally well written but some of the results are not conclusive. It is not convincing that only one Mongu AERONET aerosol model is used for this study. Just because the time-series is correlated with UV index and total column AOD it does not

mean that they can serve as proxies for AAC load when retrievals are not available. I suggest removing that conclusion from the abstract. The level 1 to level 2 pixel to cell size appears to be done arbitrarily without justification (may be calculating signal to noise ratio will help). The two step cloud masking approach for SeaWIFS is again filled with uncertainties. Changing these thresholds even slightly may alter results. The satellite/4STAR comparisons indicate that the individual level comparisons are noisy and of course the granule level averaged comparisons are better. While the discussion to the differences are explained more from primarily a statistical point it will be interesting to obtain some definitive answers on why these discrepancies exist. The authors themselves conclude that the AAC algorithm only performs "roughly" within expectations. The validation data appears to be sparse.

In conclusion while this paper is well written provides an update to the AAC algorithm, the validation portions of the paper appear to be preliminary.

---

## Referee Comment (RC3) · Anonymous Referee #3 · 23 Apr 2019

This paper provides an update on a previously published algorithm for retrieving the optical thickness of aerosols (AOT) above clouds from any one of the three satellite instruments, MODIS, VIIRS and SeaWiFS. The updated algorithm is then compared with co-located measurements of AOT above clouds from the ORACLES field campaign. Furthermore, the comparison of the long-term time-series of retrievals from both MODIS instruments currently in orbit and the VIIRS and SeaWiFS instruments. The paper provides a significant update to the previously published algorithm, as well as a valuable validation study. The paper is long and comprehensive. It certainly deserves publication in AMT, however, I think that some of the key results of the study are obscured by the lengthy discussion and I encourage the authors to take steps to elevate some of the key results, in particular the quantitative results of the comparison with the aircraft data, which are buried in the middle of the long paper.

The results of the comparison with the ORACLES data in the abstract and conclusion are presented as "performance generally in line with theoretical expectations". However, the quantitative meaning of "theoretical expectations" is obscured by the lengthy discussion. Likewise, the many sources of uncertainty and assumptions made in the algorithm are documented in great detail, but their quantitative impact on the retrieval does not clearly come through. Nevertheless, one presumes that these determine what the theoretical expectations are for the retrieval error.

Figure 7 indicates that the error statistics for the different satellite instruments are substantially different, but the understanding for what accounts for those differences does not come through in the discussion. For example, differences in resolution among the instruments are mentioned and the confounding problem of broken cloud scenes is discussed, which would presumably impact instruments with different resolutions in a different manner, however these matters are not discussed in the context of the actual errors and biases presented in the paper.

The paper would be significantly improved if the abstract and conclusions provided a more quantitative summary of the main results of the paper: the comparison with the field campaign data, rather than simply referring to the expected error. Section 2 of the paper should then make clear in a quantitative manner how the details of the retrieval scheme contribute to the error of the result. In some places in section 2 there are numbers cited for contributions to error, however in other places there are only qualitative, e.g. "results are only weakly sensitive to the value of this threshold". It would be much easier to understand what the expected error means if section 2 included a summary section and summary table that assembles the quantitative details that are currently obscured and spread across several pages. If adding such a section,

the authors should seek to reduce discussion that is not directly pertinent to evaluating the performance of the retrieval as the manuscript in current form is very lengthy.

The conclusion section includes lengthy discussion of other retrieval approaches and sensors that were not elements of the work presented earlier in the paper. The key paragraph summarizing the take-away message of the paper seems to be the second-to-last paragraph beginning line 11 on page 41 where it is noted that sensor calibration and assumed aerosol/cloud optical properties and structures are key areas to focus on for improving the retrieval. This should be the emphasis of the discussion rather than the unrelated matters of other sensors and techniques. This conclusion implies that there has been some effort to evaluate the sensitivity of the retrieval to the assumptions regarding the cloud and aerosols properties in the forward radiative transfer calculations. Again, this is not made clear in section 2. If the top-line result in the conclusion is the importance of aerosol/cloud properties and sensor calibration, then section 2 of the paper should include the quantitative support for this conclusion and this should be elevated in the abstract in conclusion sections rather than obscured by lengthy discussion of matters unrelated to the retrieval technique presented in the paper.

---

## Author Comment (AC1) · 9 May 2019

We thank the three reviewers for their useful comments and kind words about our study; below, reviewer comments are in **bold**, and our response in regular type.

In addition to changes related to the below, we have made the following additional edits in this version of the manuscript:

- Corrected affiliations for co-author Shinozuka.
- Added DOI references for ORACLES data to Data Availability section.

We have included a marked-up version of the revision using latexdiff script, which shows deletions in **red** and additions in **blue**, relative to the original manuscript.

**Reviewer 1**

Main comments (1) This paper presents a unique 20-year time series of AOD of aerosols-above-clouds in Southern Africa, with an extensive validation. A number of different satellite instruments are used to create a long time series of AAC AOD, which is useful for climate analysis. Especially the validation has been performed extensively and carefully. Figure 7 shows comprehensively the validation of the results. This interesting and important paper deserves publication. The paper is a pleasure to read: it is well written and has a clear structure. The figures are clear and informative. There is a good introduction with good referencing. The methods that are used are sound. The discussion includes the important aspect of calibration, which is different for each instrument used.

We are very happy to hear the reviewer's positive impression of our work.

**(2) In the optimal estimation retrieval scheme which is used, also the error on the AOD is retrieved. It would be informative to give with the AOD map also the AOD retrieval error map, e.g. in Figure 15.**

We have added the estimated AOD uncertainty information into the revised Figure 4. We felt that since it is highly pixel-dependent, it would be more illustrative to add it to this single-granule example than the multiannual composite shown in Figure 15. This also ties in to the discussion in Section 2.3.6., which we have now expanded as a result. Additionally, there is no accepted framework at present for propagating level 2 AOD retrieval uncertainties to longer-term aggregates such as are shown in Figure 15.

**(3) Several satellites are used with different overpass times. Is there an effect of the satellite overpass time, because of diurnal variation of the AAC AOD?**

It's possible, although far from sources we think this is unlikely to be significant (since the lifetime of the aged smoke can be several days). There's no easy way to tell from these instruments; geostationary data might help if the variation is larger than retrieval uncertainty. We've added some text to Section 2.1 on this topic at the point where overpass times are discussed.

**Caption Figure 2: To which period do the data refer? Please correct: \unit nm.**

This is 2002-2015 (due to availability of the fire product); the caption has been updated and corrected as indicated.

**p. 6, l. 26: E\_lambda should be defined perpendicular to the solar direction, and at 1 AU. In principle, E\_lambda should hold at exactly the same time as the radiance measurement**

We've added this note to the text here.

**Eqs. 7-8 on p. 11, and text below the equations: For these brightness tests, the reflectance becomes a sun-normalized radiance. Does this translation from reflectance to radiance only hold for specific SZA range? Please give the angular range.**

We've added the typical solar angle range sampled by SeaWiFS over the study region. The thresholds were optimised based on this region so might not hold outside it; we've added a note to that effect.

**p. 13, l. 10: superfluous bracket after 10 m.**

Thanks; we have corrected this.

**p. 13, l. 14: do you also correct for NO2, which absorbs in the blue range ?**

We do not, due to lack of a reanalysis which can be used in a similar way as for O3/H2O. This is a simplification often made in AOD retrieval algorithms for this reason. Additionally, since NO2 has a short lifetime it is highest near sources and often in the boundary layer, so in practical terms likely has a smaller effect on far-from-source, above-cloud cases such as often found in this study. We've added a few sentences and reference (Ahmad et al., 2007) discussing this in Section 2.3.3.

**p. 16, l. 8: then > than**

We cannot find the word "then" at this location or nearby, so perhaps the reviewer noted the wrong line/page here. The production office should catch this if the paper is eventually accepted for publication.

**Figure 4: at which wavelength does this AOD hold?**

At 550 nm. As noted in the text, mentions of AOD without a wavelength refer to 550 nm. However we have now added it in to the caption for clarity as well.

**p. 22, I. 2: Please give a reference for these main metrics.**

We are not sure what references are needed here, since these are all standard statistical metrics. In the revision we've added a note that these metrics are often used for evaluation of AOD data sets, and citations to other Deep Blue papers, to show that this sort of analysis is how we have looked at Deep Blue evaluation before.

**p. 22, l. 18: please give an equation for f, in which it is related to S in Eq. 2.**

Although this calculation was described fully in the text, in the revised version we've added notation and the relevant inequality that gets checked at this point, including the relation of the uncertainty to Equation 2.

**p. 37, l. 11: featured > feature**

Thanks; we have corrected this.

**Figure 15: Multiannual is too vague. Please give the time period for which these data hold. You could give this information in a table, together with the overpass times of the various satellites.**

It is sensor-dependent; we have added the relevant years to the caption. Overpass times were already given and discussed in Section 2.1.

**p. 39, l. 30: incorporation**

Thanks; we have corrected this.

**p. 40, l. 8-9: could these differences be due to different periods used, or different time of day?**

Based on the overlapping years (shown in Figure 13) and the fact that SeaWiFS's time was interim between Terra and Aqua, while SeaWiFS is generally higher, we think calibration likely dominates. We've added a note later on page 40 (when talking about potential future applications to geostationary data) to remind the reader that diurnal variations can't be assessed using only the Sun-synchronous sensors we have here.

**p. 40, l. 13-14: on board calibration using the sun and lamps is used for GOME type sensors, OMI, etc.**

Right, but our point here is a common external reference which multiple sensors can use; we've expanded the sentence here to make this clearer (added "against a common reference source"). To be clear, all the instruments we use have on-board calibration, the advantage of a common external reference is to calibrate against a common standard so the baseline is the same.

**Reviewer 2**

The goal of the paper by Sayer et al is to provide an update of the aerosol above cloud (AAC) algorithm that retrieves above cloud AOD and liquid COD. This algorithm can then be applied both over land and ocean for sensors such as SeaWiFS, MODIS, and MODIS. The paper also evaluates the results of the algorithm from 2016 and 2017 from the (4STAR/HSRL2) ORACLES filed campaign. The paper is generally well written but some of the results are not conclusive.

We are happy that the reviewer finds the study well-written. Yes, some of the results are not conclusive – we are careful to be honest and thorough about what we can and can't say, as we don't want to exaggerate the state of the science. This is an emerging area of research with limited available validation data (more or less the SAFARI-2000 campaign, which we also used, plus the contents of the current ACP-AMT Special Issue to which this study belongs), so this and other algorithms to quantify AACs are not yet operational and still in research mode. We have tried to make this point clear in particular through the Introduction and Conclusions, and feel we have only made claims which we can support, and identified what is needed to move forwards.

**It is not convincing that only one Mongu AERONET aerosol model is used for this study.**

In this paper we explained the rationale for the choice of optical model, and in the previous study (Sayer et al 2016) there's an analysis quantifying this assumption's impact on the retrievals. In the present study we provide additional examination of the single scattering albedo (SSA) assumption based on diverse measurements from

this region. Our previous study showed that SSA was the most critical assumption in the optical model, but that it was a second-order effect overall. Our SSA discussion and Conclusions to this paper note that updates to the optical model may improve results but that there's limited observations to tune to. This is one of many reasons we recommend future detailed characterisation of these phenomena from airborne observations. In the revision, we've mentioned the optical model more explicity in the conclusions. We have also added a reference to Pistone *et al.* (2019), which was not finalised at the time our original submission was being prepared, but uses airborne ORACLES 2016 measurements of SSA and shows that our assumption is reasonable. Combined with the slimming-down of some material suggested by Reviewer 3, we hope that some of these points will come across more clearly in the revised version.

**Just because the time-series is correlated with UV index and total column AOD it does not mean that they can serve as proxies for AAC load when retrievals are not available. I suggest removing that conclusion from the abstract.**

We have removed this from the Abstract, as requested; we were not intending to state it's a perfect relationship, but simply note that from these data sets it's quite a strong one. However it's not one of the main points of our study, so we're happy to remove it from the Abstract. We also changed the body text to refer to it as a proxy for variations in AAC loading, rather than a proxy for AAC loading.

**The level 1 to level 2 pixel to cell size appears to be done arbitrarily without justification (may be calculating signal to noise ratio will help).**

As stated in Section 2.1, the justification is to provide a data product at the same spatial resolution as the existing Deep Blue products, because the goal is eventually to incorporate this AAC retrieval algorithm within those products. For MODIS and VIIRS it is also related to sensor design (full and half scan dimensions, respectively). If one were doing something stand-alone, one might make different choices, but that is not the intention here. Signal to noise is not the main factor as retrieval uncertainty is mostly algorithmic rather than radiometric. We've expanded the end of section 2.1 to provide more background for the decision here.

**The two step cloud masking approach for SeaWIFS is again filled with uncertainties. Changing these thresholds even slightly may alter results.**

That is true of any cloud mask and unfortunately this is due to the design and development of the SeaWiFS mission. As noted in the text, SeaWiFS did not have a standard cloud mask product because (unlike MODIS or VIIRS) it wasn't used to routinely process cloud retrievals. SeaWiFS was mostly used for ocean colour retrievals over ocean, and NDVI mapping over land, and the cloud masks used there were designed to filter out thin and thick clouds as well as aerosols, and pixels near to those features, and were quite clear-sky conservative. So they are not useful to this type of AAC retrieval we have to create our own because there's no alternative. We directly state in the text the limitations of this compared to the text, and in the revision we provide a bit more background to historical SeaWiFS cloud masking, and mention this in the Conclusions as well.

The satellite/4STAR comparisons indicate that the individual level comparisons are noisy and of course the granule level averaged comparisons are better. While the discussion to the differences are explained more from primarily a statistical point it will be interesting to obtain some definitive answers on why these discrepancies exist. The authors themselves conclude that the AAC algorithm only performs "roughly" within expectations. The validation data appears to be sparse. In conclusion while this paper

**is well written provides an update to the AAC algorithm, the validation portions of the paper appear to be preliminary.**

We would love to know but since validation is an inherently statistical exercise and retrievals are underdetermined it's difficult (or impossible) to be definitive. The retrievals provide an uncertainty estimate (measure of range of plausible solutions); the airborne AOD data have nonzero uncertainty, and we simply don't have measurements of all the other relevant quantities (e.g. aerosol size distribution, cloud structure and particle size) with high accuracy to use as truth to interrogate individual matchups. As a result, what we can say is whether or not the retrievals are consistent with our expectations, given what we do and do not know about the true state. In the revised version, we have mentioned this again within the Conclusions, as well as the Abstract. The rationale for providing both individual and granule statistics was, as stated in the text, to ameliorate the effects of sampling frequency differences between different flight legs.

Performing "roughly within expectations" is exactly what one would hope for as it means that your expectations were correct. As noted earlier, we are deliberately not trying to over-hype or over-sell our results and claim that the problem of AAC retrieval is solved because clearly with the satellite instrumentation we have to date, uncertainties remain large. Also, as discussed earlier, one of the limits is that there simply haven't been many measurements of these phenomena which can be used for validation, which is why the Conclusion (and in the revised version, also the Abstract) calls for more. We hope we have made this point sufficiently clear through the paper and would note in response to this comment that Reviewer 1 felt that "Especially the validation has been performed extensively and carefully" and reviewer 3 described it as "comprehensive" (albeit with some parts obscured by the length).

**Reviewer 3**

This paper provides an update on a previously published algorithm for retrieving the optical thickness of aerosols (AOT) above clouds from any one of the three satellite instruments, MODIS, VIIRS and SeaWiFS. The updated algorithm is then compared with co-located measurements of AOT above clouds from the ORACLES field campaign. Furthermore, the comparison of the long-term time-series of retrievals from both MODIS instruments currently in orbit and the VIIRS and SeaWiFS instruments. The paper provides a significant update to the previously published algorithm, as well as a valuable validation study. The paper is long and comprehensive. It certainly deserves publication in AMT, however, I think that some of the key results of the study are obscured by the lengthy discussion and I encourage the authors to take steps to elevate some of the key results, in particular the quantitative results of the comparison with the aircraft data, which are buried in the middle of the long paper.

We are pleased that the reviewer sees the merit in our study.

The results of the comparison with the ORACLES data in the abstract and conclusion are presented as "performance generally in line with theoretical expectations". However, the quantitative meaning of "theoretical expectations" is obscured by the lengthy discussion. Likewise, the many sources of uncertainty and assumptions made in the algorithm are documented in great detail, but their quantitative impact on the retrieval does not clearly come through. Nevertheless, one presumes that these determine what the theoretical expectations are for the retrieval error. Figure 7 indicates that the error statistics for the different satellite instruments are substantially different, but the understanding for what accounts for those differences does not come through in the discussion. For example, differences in resolution among the instruments are mentioned and the confounding problem of broken cloud scenes is discussed, which would presumably impact instruments with different resolutions in a different manner, however these matters are not discussed in the context of the actual errors and biases presented in the paper.

The paper would be significantly improved if the abstract and conclusions provided a more quantitative summary of the main results of the paper: the comparison with the field campaign data, rather than simply referring to the expected error. Section 2 of the paper should then make clear in a quantitative manner how the details of the retrieval scheme contribute to the error of the result. In some places in section 2 there are numbers cited for contributions to error, however in other places there are only qualitative, e.g. "results are only weakly sensitive to the value of this threshold". It would be much easier to understand what the expected error means if section 2 included a summary section and summary table that assembles the quantitative details that are currently obscured and spread across several pages. If adding such a section the authors should seek to reduce discussion that is not directly pertinent to evaluating the performance of the retrieval as the manuscript in current form is very lengthy. The conclusion section includes lengthy discussion of other retrieval approaches and sensors that were not elements of the work presented earlier in the paper. The key paragraph summarizing the take-away message of the paper seems to be the second-to-last paragraph beginning line 11 on page 41 where it is noted that sensor calibration and assumed aerosol/cloud optical properties and structures are key areas to focus on for improving the retrieval. This should be the emphasis of the discussion rather than the unrelated matters of other sensors and techniques. This conclusion implies that there has been some effort to evaluate the sensitivity of the retrieval to the assumptions regarding the cloud and aerosols properties in the forward radiative transfer calculations. Again, this is not made clear in section 2. If the top-line result in the conclusion is the importance of aerosol/cloud properties and sensor calibration, then section 2 of the paper should include the quantitative support for this conclusion and this should be elevated in the abstract in conclusion sections rather than obscured by lengthy discussion of matters unrelated to the retrieval technique presented in the paper.

We interpret the reviewer's comments as requests to:

1. Increase the signposting to quantitative results throughout the paper, both in the theoretical uncertainty and in the comparison with airborne data.

2. Remove background details not directly related to algorithm development/performance to decrease the overall length of the manuscript and improve readability.

There is a slight difficulty here in that reviewer 2 (and to a lesser extent reviewer 1) requested additions to the text, including some extra background material, while reviewer 3 requests a few additions but more cutting out material from the text that isn't directly about algorithm specifics. However we do agree that trimming things down could improve overall clarity. The additions requested by reviewers 1 and 2 add about 2 pages (from 50 to 52 pages in submission draft format) to the overall length, compared to the initial submission of the paper. In the revision, we have attempted to satisfy all three reviewers, and have made changes to account for the two main thrusts of Reviewer 3's comments as well:

1. We have added a table (new Table 2) summarising the contributions to the retrieval error budget from various sources. These are framed in terms of TOA reflectance because the effect on the retrieval itself is nonlinear and a function of conditions. So it would be misleading to pin down a number and say that X% uncertainty on some

factor leads to Y% uncertainty on AOT. This is a strength of the Optimal Estimation framework, which is what we refer to when we say the theoretical expectations. As we note in the text, we are also cautious about overinterpreting the comparison results quantitatively because the differences between deployments indicates that the available sampling of validation data so far may not be sufficient to give a robust quantification. We also feel it's important to point out that since the uncertainties are so variable, framing them in terms of the expected uncertainty (which is shown quantitatively in several plots) is probably the most sensible approach. However we acknowledge that the discussion is long and some things can get buried. We have rewritten the paper at several places (both the Abstract and Conclusions, as well as at points throughout the paper) to make these points clearer and signpost them. Note that Figure 4 also now contains a map of the estimated uncertainty, at the request of Reviewer 1, which also helps bring a quantitative illustration for this one specific example.

2. We have been through the manuscript to trim out text where we can so that the quantitative results feel less buried. There has been editing throughout and we paid particular attention to the Abstract and Conclusion that the reviewer identified. The careful edits out made in response to Reviewer 3's comments have contributed to decreasing the overall length from 52 to 50 pages in draft format (despite the added content). Thus, while the overall length is the paper, we have added the material reviewers asked for while streamlining and better signposting the content to improve readability. We hope this will be satisfactory.